# Masked graph modeling for molecule generation

Omar Mahmood [1], Elman Mansimov[2], Richard Bonneau [3] & Kyunghyun Cho [2✉]

De novo, in-silico design of molecules is a challenging problem with applications in drug discovery and material design. We introduce a masked graph model, which learns a distribution over graphs by capturing conditional distributions over unobserved nodes (atoms) and edges (bonds) given observed ones. We train and then sample from our model by iteratively masking and replacing different parts of initialized graphs. We evaluate our approach on the QM9 and ChEMBL datasets using the GuacaMol distribution-learning benchmark. We find that validity, KL-divergence and Fréchet ChemNet Distance scores are anti-correlated with novelty, and that we can trade off between these metrics more effectively than existing models. On distributional metrics, our model outperforms previously proposed graph-based approaches and is competitive with SMILES-based approaches. Finally, we show our model generates molecules with desired values of specified properties while maintaining physiochemical similarity to the training distribution.

[1] Center for Data Science, New York University, New York, NY, USA. [2] Department of Computer Science, Courant Institute of Mathematical Sciences, New York, NY, USA. [3] Center for Genomics and Systems Biology, New York University, New York, NY, USA. ✉email: kyunghyun.cho@nyu.edu

The design of de novo molecules in-silico with desired properties is an essential part of drug discovery and materials design but remains a challenging problem due to the very large combinatorial space of all possible synthesizable molecules[1]. Recently, various deep generative models for the task of molecular graph generation have been proposed, including: neural autoregressive models[2,3], variational autoencoders[4,5], adversarial autoencoders[6], and generative adversarial networks[7,8].

A unifying theme behind these approaches is that they model the underlying distribution $p^\star(G)$ of molecular graphs $G$. Once the underlying distribution is captured, new molecular graphs are sampled accordingly. As we do not have access to this underlying distribution, it is typical to explicitly model $p^\star(G)$ by a distribution $p_\theta(G)$. This is done using a function $f_\theta$ so that $p_\theta(G) = f_\theta(G)$. The parameters $\theta$ are then learned by minimizing the KL-divergence $KL(p^\star||p_\theta)$ between the true distribution and the parameterized distribution. Since we do not have access to $p^\star(G)$, $KL(p^\star||p_\theta)$ is approximated using a training set $D = (G_1, G_2, \ldots, G_M)$ which consists of samples from $p^\star$. Once the model has been trained on this distribution, it is used to carry out generation.

Each of these approaches makes unique assumptions about the underlying probabilistic structure of a molecular graph. Autoregressive models[2,3,9–12] specify an ordering of atoms and bonds in advance to model the graph. They decompose the distribution $p(G)$ as a product of temporal conditional distributions $p(g_t|G_{<t})$, where $g_t$ is the vertex or edge to be added to $G$ at time $t$ and $G_{<t}$ are the vertices and edges that have been added in previous steps. Generation from an autoregressive model is often done sequentially by ancestral sampling. Defining such a distribution requires fixing an ordering of the nodes and vertices of a graph in advance. Although directed acyclic graphs have canonical orderings based on breadth-first search (BFS) and depth-first search (DFS), graphs can take a variety of valid orderings. The choice of ordering is largely arbitrary, and it is hard to predict how a particular choice of ordering will impact the learning process[13].

Latent variable models such as variational autoencoders and adversarial autoencoders assume the existence of unobserved (latent) variables $Z = \{z_1, z_2, \ldots, z_k\}$ that aim to capture dependencies among the vertices $V$ and edges $E$ of a graph $G$. Unlike an autoregressive model, a latent variable model does not necessarily require a predefined ordering of the graph[14]. The generation process consists of first sampling latent variables according to their prior distributions, followed by sampling vertices and edges conditioned on these latent variable samples. However, learning the parameters $\theta$ of a latent variable model is more challenging than learning the parameters of an autoregressive model. It requires marginalizing latent variables to compute the marginal probability of a graph, i.e., $p(G) = \int_Z p(G|Z)p(Z)dZ$, which is often intractable. Recent approaches have focused on deriving a tractable lower-bound to the marginal probability by introducing an approximate posterior distribution $q(Z)$ and maximizing this lowerbound instead[4–6]. Unlike variational autoencoders, generative adversarial networks (GAN) do not use KL-divergence to measure the discrepancy between the model distribution and data distribution and instead estimate the divergence as a part of learning.

In addition to using a specific factorization, each model uses a specific representation of molecules; two such representations are string-based and graph-based. The ability of a language model to model molecules is limited by the string representation used[15]. Directly modeling molecules as graphs bypasses the need to find better ways of serializing molecules as strings. It also allows for the use of graph-specific features such as distances between atoms, which are not readily encoded as strings. Developing datasets and benchmarks that incorporate these features would enable more informative comparisons between models that use different molecular representations.

Existing graph-based generative models of molecules attempt to directly model the joint distribution. Some of these models follow the autoregressive framework earlier described. Li et al.[16] proposed a deep generative model of graphs that predicts a sequence of transformation operations to generate a graph. You et al.[17] proposed an RNN-based autoregressive generative model that generates components of a graph in breadth-first search (BFS) ordering. To speed up the autoregressive graph generation and improve scalability, Liao et al.[18] extended autoregressive models of graphs by adding blockwise parallel generation. Dai et al.[19] proposed an autoregressive generative model of graphs that utilizes sparsity to avoid generating the full adjacency matrix and generates novel graphs in log-linear time complexity. Grover et al.[20] proposed a VAE-based iterative generative model for small graphs. They restrict themselves to modeling only the graph structure, not a full graph including node and edge features for molecule generation. Liu et al.[21] proposed a graph neural network model based on normalizing flows for memory-efficient prediction and generation. Mercado et al.[22] proposed a graph neural network-based generative model that learns functions corresponding to whether to add a node to a graph, connect two existing nodes or terminate generation. These learned functions are then used to generate de-novo graphs. The approach requires selecting an ordering of graph components, which the authors choose to be the BFS ordering.

There are also latent variable methods for graph generation. For example, Simonovsky and Komodakis[23] proposed a graph VAE to generate graph representations of molecules. Jin et al.[24] proposed using a VAE to generate a junction tree followed by the generation of the molecule itself. This approach is likely to generate valid chemical structures as it uses a predetermined vocabulary of valid molecular substructures. Kwon et al.[25] proposed a non-autoregressive graph variational autoencoder, which is trained with additional learning objectives to the standard VAE ELBO for unconditional and conditional molecular graph generation.

Along with these works on autoregressive and latent variable generative models of graphs, there is work applying reinforcement learning objectives to the task of molecular graph generation[26–28] and reaction-driven molecule design[29–31]. In addition, Yang et al.[32] proposed a target augmentation approach for improving molecular optimization, a model-agnostic framework that can be used with any black box model. Hence several existing works on generating graph representations of molecules (see the Supplementary Discussion section of the Supplementary Information for more examples) directly model the joint distribution $p(G)$ or incorporate additional objectives that can be used with a variety of models.

Here, we explore another approach to probabilistic graph generation based on the insight that we do not need to model the joint distribution $p(G)$ directly to be able to sample from it. We propose a masked graph model, a generative model of graphs that learns the conditional distribution of masked graph components given the rest of the graph, induced by the underlying joint distribution. This allows us to use a procedure similar to Gibbs sampling to generate new molecular graphs, as Gibbs sampling requires access only to conditional distributions. Concretely, our approach, to which we refer as *masked graph modeling*, parameterizes and learns conditional distributions $p(\eta|G_{\setminus\eta})$ where $\eta$ is a subset of the components (nodes and edges) of $G$ and $G_{\setminus\eta}$ is a graph without those components (or equivalently with those components masked out). With these conditional distributions estimated from data, we sample a graph by iteratively updating its components. At each generation iteration, this involves choosing a subset of components, masking them, and sampling new values

for them according to the corresponding conditional distribution. By using conditional distributions, we circumvent the assumptions made by previous approaches to model the unconditional distribution. We do not need to specify an arbitrary order of graph components, unlike in autoregressive models, and learning is exact, unlike in latent variable models. Our approach is inspired by masked language models[33] that model the conditional distribution of masked words given the rest of a sentence, which have shown to be successful in natural language understanding tasks[34–39] and text generation[40]. As shown in previous works[40–42], sampling from a trained denoising autoencoder, which is analogous to sampling from our masked graph model, is theoretically equivalent to sampling from the full joint distribution. Therefore, even though we train our model on conditional distributions, sampling repeatedly from these distributions is equivalent to sampling from the full joint distribution of graphs.

In this work, we evaluate our approach on two popular molecular graph datasets, QM9[43,44] and ChEMBL[45], using a set of five distribution-learning metrics introduced in the GuacaMol benchmark[46]: the validity, uniqueness, novelty, KL-divergence[47] (KLD) and Fréchet ChemNet Distance[48] (FCD) scores. The KL-divergence and Fréchet ChemNet Distance scores are measures of the similarity between generated molecules and molecules from the combined training, validation and test distributions, which we call the dataset distribution. We find that the validity, Fréchet ChemNet Distance and KL-divergence scores are highly correlated with each other and inversely correlated with the novelty score. We show that state-of-the-art autoregressive models are ineffective in controlling the trade-off between novelty and the validity, Fréchet ChemNet Distance, and KL-divergence scores, whereas our masked graph model provides effective control over this trade-off. Overall, the proposed masked graph model, trained on the graph representations of molecules, outperforms previously proposed graph-based generative models of molecules and performs comparably to several SMILES-based models. Additionally, our model achieves comparable performance on validity, uniqueness, and KL-divergence scores compared to state-of-the-art autoregressive SMILES-based models, but with lower Fréchet ChemNet Distance scores. We also carry out conditional generation to obtain molecules with target values of specified physiochemical properties. This involves predicting the masked out components of a molecular graph given the rest of the graph, conditioned on the whole graph having a specified value of the physiochemical property of interest. Example target properties for this approach include the LogP measure of lipophilicity, and molecular weight. We find that our model produces molecules with values close to the target values of these properties without compromising other metrics. Compared with a baseline graph generation approach, the generated molecules maintain physiochemical similarity to the training distribution even as they are optimized for the specified metric. Finally, we find that our method is computationally efficient, needing little time to generate new molecules.

## Results

**Masked graph modeling overview**. A masked graph model (MGM) operates on a graph $G$, which consists of a set of $N$ vertices $\mathcal{V} = \{v_i\}_{i=1}^N$ and a set of edges $\mathcal{E} = \{e_{i,j}\}_{i,j=1}^N$. A vertex is denoted by $v_i = (i, t_i)$, where $i$ is the unique index assigned to it, and $t_i \in C_v = \{1, \ldots, T\}$ is its type, with $T$ the number of node types. An edge is denoted by $e_{i,j} = (i, j, r_{i,j})$, where $i, j$ are the indices to the incidental vertices of this edge and $r_{i,j} \in C_e = \{1, \ldots, R\}$ is the type of this edge, with $R$ the number of edge types.

We use a single graph neural network to parameterize any conditional distribution induced by a given graph. We assume

that the missing components $\eta$ of the conditional distribution $p(\eta|G_{\backslash\eta})$ are conditionally independent of each other given $G_{\backslash\eta}$:

$$p(\eta|G_{\backslash\eta}) = \prod_{v\in\mathcal{V}} p(v|G_{\backslash\eta}) \prod_{e\in\mathcal{E}} p(e|G_{\backslash\eta}), \qquad (1)$$

where $\mathcal{V}$ and $\mathcal{E}$ are the sets of all vertices and all edges in $\eta$ respectively.

To train the model, we use fully observed graphs from a training dataset $D$. We corrupt each graph $G$ with a corruption process $C(G_{\backslash\eta}|G)$, i.e. $G_{\backslash\eta} \sim C(G_{\backslash\eta}|G)$. In this work, following the work of Devlin et al.[33] for language models, we randomly replace some of the node and edge features with the special symbol MASK. After passing $G_{\backslash\eta}$ through our model we obtain the conditional distribution $p(\eta|G_{\backslash\eta})$. We then maximize the log probability $\log p(\eta|G_{\backslash\eta})$ of the masked components $\eta$ given the rest of the graph $G_{\backslash\eta}$. This is analogous to a masked language model[33], which predicts the masked words given the corrupted version of a sentence. This results in the following optimization problem:

$$\arg\max_{\theta} \mathbb{E}_{G\sim D}\mathbb{E}_{G_{\backslash\eta}\sim C(G_{\backslash\eta}|G)}\log p_{\theta}(\eta|G_{\backslash\eta}). \qquad (2)$$

Once we have trained the model, we use it to carry out generation. To begin generation, we initialize a molecule in one of two ways, corresponding to different levels of entropy. The first way, which we call training initialization, uses a random graph from the training data as an initial graph. The second way, which we call marginal initialization, initializes each graph component according to a categorical distribution over the values that component takes in our training set. For example, the probability of an edge having type $r \in C_e$ is equal to the fraction of edges in the training set of type $r$.

We then use an approach motivated by Gibbs sampling to update graph components iteratively from the learned conditional distributions. At each generation step, we sample uniformly at random a fraction $\alpha$ of components $\eta$ in the graph and replace the values of these components with the MASK symbol. We compute the conditional distribution $p(\eta|G_{\backslash\eta})$ by passing the partially masked graph through the model, sampling new values of the masked components according to the predicted distribution, and placing these values in the graph. We repeat this procedure for a total of $K$ steps, where $K$ is a hyperparameter. A schematic of this procedure is given in Supplementary Figure 4.

We carry out conditional generation using a modified version of this approach. We frame this task as generating molecules with a target value of a given physiochemical property. We use the same training and generation procedures as for unconditional generation but with an additional, conditioning, input to the model. This input $y$ is the molecule's graph-level property of interest. During training, $y$ corresponds to the ground-truth value $y^{\star}$ of the molecule's graph-level property of interest. This results in a modified version of Equation (2):

$$\arg\max_{\theta} \mathbb{E}_{G\sim D}\mathbb{E}_{G_{\backslash\eta}\sim C(G_{\backslash\eta}|G)}\log p_{\theta}(\eta|G_{\backslash\eta}, y = y^{\star}) \qquad (3)$$

During generation, $y$ instead corresponds to the target value $\hat{y}$ of this property. The initialization process is the same as for unconditional generation. Iterative sampling involves updating the graph by computing the conditional distribution $p(\eta|G_{\backslash\eta}, y = \hat{y})$.

**Mutual dependence of metrics from GuacaMol**. We evaluate our model and baseline molecular generation models on unconditional molecular generation using the distribution-learning benchmark from the GuacaMol[46] framework. We first attempt to determine whether dependence exists between metrics from the Guacamol framework. We do this because we notice

**Table 1 Spearman's correlation coefficient between benchmark metrics for results using the masked graph model on the QM9 dataset.**

|              | Validity | Uniqueness | Novelty | KL Div | Fréchet Dist |
|--------------|----------|------------|---------|--------|--------------|
| Validity     | 1.00     | −0.56      | −0.83   | 0.73   | 0.75         |
| Uniqueness   | −0.56    | 1.00       | 0.50    | −0.32  | −0.37        |
| Novelty      | −0.83    | 0.50       | 1.00    | −0.94  | −0.95        |
| KL Div       | 0.73     | −0.32      | −0.94   | 1.00   | 0.99         |
| Fréchet Dist | 0.75     | −0.37      | −0.95   | 0.99   | 1.00         |

**Table 2 Spearman's correlation coefficient between benchmark metrics for results using LSTM, Transformer Small and Transformer Regular on the QM9 dataset.**

|              | Validity | Uniqueness | Novelty | KL Div | Fréchet Dist |
|--------------|----------|------------|---------|--------|--------------|
| Validity     | 1.00     | 0.03       | −0.99   | 0.98   | 0.98         |
| Uniqueness   | 0.03     | 1.00       | 0.00    | 0.03   | 0.03         |
| Novelty      | −0.99    | 0.00       | 1.00    | −0.99  | −0.99        |
| KL Div       | 0.98     | 0.03       | −0.99   | 1.00   | 1.00         |
| Fréchet Dist | 0.98     | 0.03       | −0.99   | 1.00   | 1.00         |

that some of these metrics may measure similar properties. For example, the Fréchet and KL scores are both measures of similarity between generated samples and a dataset distribution. If the metrics are not mutually independent, comparing models using a straightforward measure such as the sum of the metrics may not be a reasonable strategy.

To determine how the five metrics are related to each other, we calculate pairwise the Spearman (rank) correlation between all metrics on the QM9 dataset[43,44], presented in Table 1, while varying the masking rate, initialization strategy and number of sampling iterations $K$. We carry out a similar run for three baseline autoregressive SMILES-based models that we train ourselves: two Transformer models[3] with different numbers of parameters (Transformer Small and Transformer Regular) and an LSTM. Each of these autoregressive models has a distribution output by a softmax layer over the SMILES vocabulary at each time step. We implement a sampling temperature parameter in this distribution to control its sharpness. By increasing the temperature, we decrease the sharpness, which increases the novelty. The Spearman correlation results for these baselines are shown in Table 2.

From Tables 1 and 2, we make three observations. First, the validity, KL-divergence and Fréchet Distance scores correlate highly with each other. Second, these three metrics correlate negatively with the novelty score. Finally, uniqueness does not correlate strongly with any other metric. These observations suggest that we can look at a subset of the metrics, namely the uniqueness, Fréchet and novelty scores, to gauge generation quality. We now carry out experiments to determine how well MGM and baseline models perform on the anti-correlated Fréchet and novelty scores, which are representative of four of the five evaluation metrics.

**Analysis of representative metrics**. To examine how the masked graph model and baseline autoregressive models balance the Fréchet ChemNet Distance and novelty scores, we plot these two metrics against each other in Fig. 1. To obtain the points for the masked graph models, we evaluate the scores after various numbers of generation steps. For the QM9 MGM points, we use

both training and marginal initializations, which start from the top left and bottom right of the graph respectively, and converge in between. For the ChEMBL MGM points, we use only training initialization.

On both QM9 and ChEMBL, we see that as novelty increases, the Fréchet ChemNet Distance score decreases for the masked graph models as well as for the LSTM and Transformer models. We also see that the line's slope, which represents the marginal change in Fréchet ChemNet Distance score per unit change in novelty score, has a lower magnitude for the masked graph model than for the autoregressive models. This shows that our model trades off novelty for similarity to the dataset distributions (as measured by the Fréchet score) more effectively relative to the baseline models. This gives us a higher degree of controllability in generating samples that are optimized towards either metric to the extent desired.

On QM9, we see that our masked graph models with a 10% or 20% masking rate maintain a larger Fréchet ChemNet Distance score as the novelty increases, compared to the LSTM and Transformer models. Several of the MGM points on the plot are beyond the Pareto frontier formed by each baseline model. On ChEMBL, the LSTM and Transformer models generally achieve a higher combination of novelty and Fréchet ChemNet Distance score than does the masked graph model with either masking rate. However, to the bottom right of Fig. 1b, we can see a few points corresponding to the 5% masking rate that are beyond the Pareto frontier of the points formed by the Transformer Regular model.

We also observe that for ChEMBL, which contains larger molecules, using a 1% masking rate yields points that are beyond the Pareto frontier of those obtained using a 5% masking rate. This further indicates that masking a large number of components hurts generation quality, even if this number represents a small percentage of the graph.

We plot validity against novelty in Supplementary Figure 3 and observe that the same analysis holds for the trade-off between these two metrics. Hence even though state-of-the-art autoregressive models can trade off between representative metrics by changing the sampling strategy, the trade-off is poor and leads to a rapid decline in molecule quality as the novelty increases. MGM, on the other hand, is able to maintain a similar molecule quality as the novelty increases.

**Comparison with baseline models**. We now compare distributional benchmark results for MGM using our 'best' initialization strategy and masking rate (see the Supplementary Discussion section of the Supplementary Information for details) to baseline models. The baseline models include models we train ourselves and those for which we obtain results from the literature. The distributional benchmark results on QM9 and ChEMBL are shown in Table 3 and Table 4 respectively.

On QM9, our model performs comparably to existing SMILES-based methods. Our approach shows higher validity and uniqueness scores compared to CharacterVAE[49] and GrammarVAE[50], while having a lower novelty score. Compared to the autoregressive LSTM and Transformer models, our model has lower validity, KL-divergence and Fréchet Distance scores; however it exhibits slightly higher uniqueness and significantly higher novelty scores.

Compared to the graph-based models, our approach performs similarly to or better than existing approaches. Our approach has higher validity and uniqueness scores compared to GraphVAE[23] and MolGAN[51], and a lower novelty score. KLD and Fréchet Distance scores are not provided for these two models. Our model outperforms the non-autoregressive graph VAE[25] on all metrics except novelty.

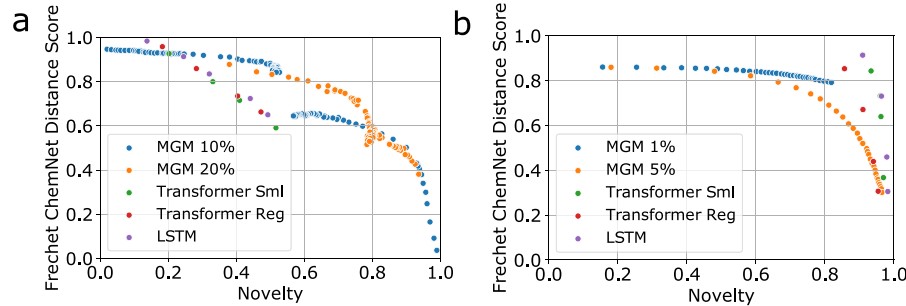

**Fig. 1 Plots of the Fréchet ChemNet Distance score against novelty on QM9 and ChEMBL.** The FCD score and novelty are two anti-correlated metrics from the GuacaMol[46] distribution-learning benchmark. Each point corresponds to the values of these two metrics for a set of molecules that are generated using the same model with the same generation hyperparameters. Different points of the same color correspond to different sets of molecules, with each set generated from the same model using different generation hyperparameters (number of generation iterations and masking rate for the masked graph models, sampling temperature for autoregressive models). The percentages indicated next to MGM in the figure legends indicate the masking rate at generation time. (For example, MGM 10% indicates an MGM model with a generation masking rate of 10%.) **a** Plots for QM9. For each QM9 MGM model, the series of points originating at the top left of the graph corresponds to training initialization, whereas the series of points originating at the bottom right corresponds to marginal initialization. **b** Plots for ChEMBL. For ChEMBL, only training initialization was used to sample valid molecules due to computational constraints, as marginal initialization did not yield enough valid molecules to calculate reliable distributional metrics in a reasonable amount of time. This is likely because the masking rate is low so it would take a long time for the sampler to converge to the training distribution. Using a high masking rate would result in a large number of spurious edges, which would be problematic for the MPNN to handle. Finding a way to alleviate this issue would be a valuable direction for future work.

**Table 3 Distributional results on QM9. CharacterVAE[49], GrammarVAE[50], GraphVAE[23] and MolGAN[51] results are taken from Cao and Kipf[51].**

|  | Model | Valid | Uniq | Novel | KL Div | Fréchet Dist |
|---|---|---|---|---|---|---|
| SMILES | CharacterVAE | 0.103 | 0.675 | 0.900 | N/A | N/A |
|  | GrammarVAE | 0.602 | 0.093 | 0.809 | N/A | N/A |
|  | LSTM (ours) | 0.980 | 0.962 | 0.138 | 0.998 | 0.984 |
|  | Transformer Sml (ours) | 0.947 | 0.963 | 0.203 | 0.987 | 0.927 |
|  | Transformer Reg (ours) | 0.965 | 0.957 | 0.183 | 0.994 | 0.958 |
| Graph | GraphVAE | 0.557 | 0.760 | 0.616 | N/A | N/A |
|  | MolGAN | 0.981 | 0.104 | 0.942 | N/A | N/A |
|  | NAT GraphVAE (ours) | 0.875 | 0.317 | 0.895 | 0.843 | 0.509 |
|  | MGM (ours proposed) | 0.886 | 0.978 | 0.518 | 0.966 | 0.842 |

NAT GraphVAE[25] stands for non-autoregressive graph VAE. Models labelled as 'ours' were trained by us and subsequently used to carry out generation. Our masked graph model results correspond to a 10% masking rate and training graph initialization, which has the highest geometric mean for all five benchmark metrics. (See the Supplementary Discussion section of the Supplementary Information for details.) Values of validity(↑), uniqueness(↑), novelty(↑), KL Div(↑) and Fréchet Dist(↑) metrics are between 0 and 1.

**Table 4 Distributional results on ChEMBL. LSTM, Graph MCTS[52], AAE[67], ORGAN[62] and VAE[49] (with a bidirectional GRU[53] as encoder and autoregressive GRU[53] as decoder) results are taken from Brown et al.[46].**

|  | Model | Valid | Uniq | Novel | KL Div | Fréchet Dist |
|---|---|---|---|---|---|---|
| SMILES | AAE | 0.822 | 1.000 | 0.998 | 0.886 | 0.529 |
|  | ORGAN | 0.379 | 0.841 | 0.687 | 0.267 | 0.000 |
|  | VAE | 0.870 | 0.999 | 0.974 | 0.982 | 0.863 |
|  | LSTM | 0.959 | 1.000 | 0.912 | 0.991 | 0.913 |
|  | Transformer Sml (ours) | 0.920 | 0.999 | 0.939 | 0.968 | 0.859 |
|  | Transformer Reg (ours) | 0.961 | 1.000 | 0.846 | 0.977 | 0.883 |
| Graph | Graph MCTS | 1.000 | 1.000 | 0.994 | 0.522 | 0.015 |
|  | NAT GraphVAE | 0.830 | 0.944 | 1.000 | 0.554 | 0.016 |
|  | MGM (ours proposed) | 0.849 | 1.000 | 0.722 | 0.987 | 0.845 |

NAT GraphVAE[25] stands for non-autoregressive graph VAE. Models labelled as 'ours' were trained by us and subsequently used to carry out generation. Our masked graph model results correspond to a 1% masking rate and training graph initialization, which has the highest geometric mean for all five benchmark metrics. (See the Supplementary Discussion section of the Supplementary Information for details.) Values of validity(↑), uniqueness(↑), novelty(↑), KL Div(↑) and Fréchet Dist(↑) metrics are between 0 and 1.

On ChEMBL, our approach outperforms existing graph-based methods. Compared to graph MCTS[52] and non-autoregressive graph VAE[25], our approach shows lower novelty scores while having significantly higher KL-divergence and Fréchet Distance scores. The baseline graph-based models do not capture the properties of the dataset distributions, as shown by their low KL-divergence scores and almost-zero Fréchet scores. This demonstrates that our proposed approach outperforms graph-based

methods in generating novel molecules that are similar to the dataset distributions.

The proposed masked graph model is competitive with models that rely on the SMILES representations of molecules. It outperforms the GAN-based model (ORGAN) across all five metrics and outperforms the adversarial autoencoder model (AAE) on all but the uniqueness score (both have the maximum possible score) and the novelty score. It performs comparably to the VAE model with an autoregressive GRU[53] decoder on all metrics except novelty. Our approach lags behind the LSTM, Transformer Small and Transformer Regular SMILES-based models on the ChEMBL dataset. It outperforms both Transformer models on KL-divergence score but

underperforms them on validity, novelty and Fréchet score. Our approach also results in lower scores across most of the metrics when compared to the LSTM model.

Some examples of generated molecules after the final sampling iteration are shown in Supplementary Figures 6 and 7. Full lists of molecules can be accessed via the Data Availability section.

**Generation trajectories**. We present a few sampling trajectories of molecules from the proposed masked graph model in Figs. 2–3. Each image represents the molecule after a certain number of sampling iterations; the first image in a figure is the molecular

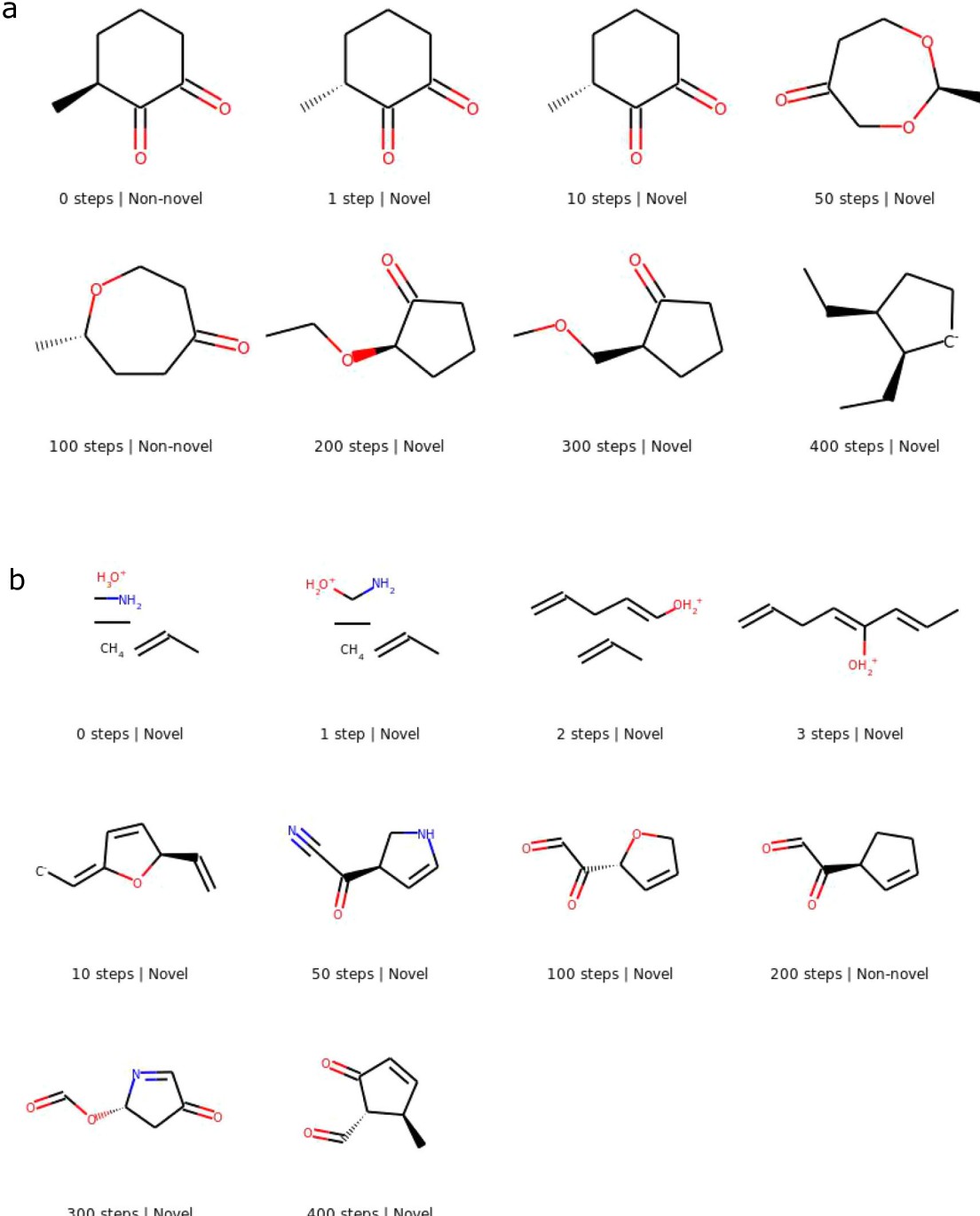

**Fig. 2 Generation trajectory of a molecule each for training initialization and marginal initialization.** The model is trained on QM9, and generation is carried out using a 10% masking rate. **a** Training initialization. **b** Marginal initialization.

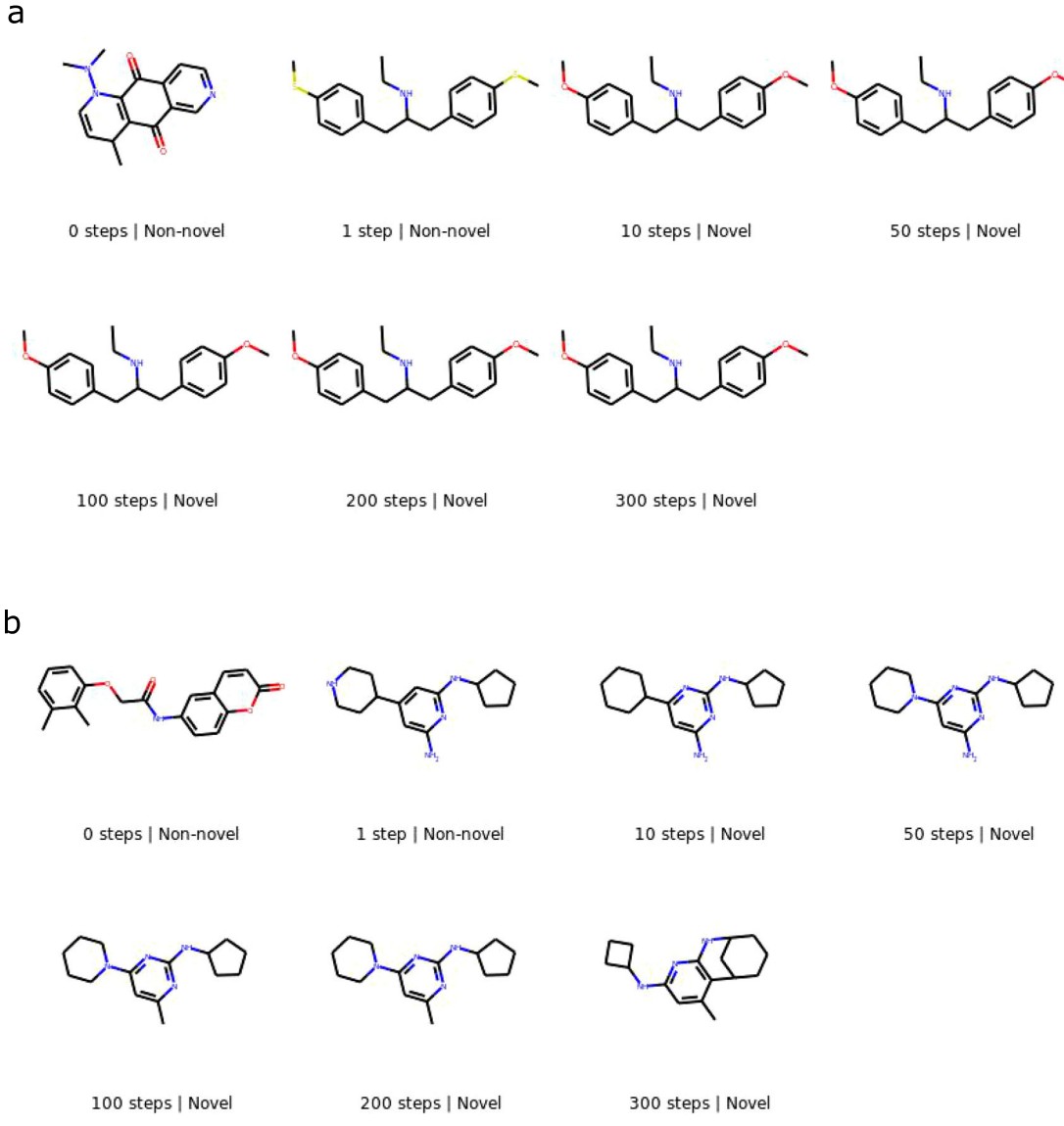

**Fig. 3 Generation trajectory of a molecule each for a 1% and a 5% masking rate.** The model is trained on ChEMBL, and generation is carried out using training initialization. **a** 1% masking rate. **b** 5% masking rate.

graph initialization before any sampling steps are taken. Figure 2 shows a trajectory each for training and marginal initializations with a 10% masking rate. Figure 3 shows a trajectory each for 1% and 5% masking rates with training initialization. All molecules displayed in the figures are valid, but molecules corresponding to some of the intermediate steps not shown may not be.

Figure 2a shows the trajectory of a molecule initialized as a molecule from the QM9 training set. As generation progresses, minor changes are made to the molecule, yielding novel molecules. After 100 generation steps, the molecule has converged to another non-novel molecule. Further generation steps yield novel molecules once again, with the molecule's structure gradually moving further away from the initialized molecule.

Figure 2b shows the trajectory of a molecule initialized from the marginal distribution of the QM9 training set. The initialized graph consists of multiple disjoint molecular fragments. Over the first three generation steps, the various nodes are connected to form a connected graph. These changes are more drastic than those in the first few steps of generation with training initialization. The molecule undergoes significant changes over the next few steps until

it forms a ring and a chiral center by the 10-th step. The molecule then evolves slowly until it converges to a non-novel molecule by 200 steps. Further generation steps yield a series of novel molecules once again.

Figure 3a shows the trajectory of a ChEMBL molecule with a 1% masking rate. In the first step, the molecule changes from one training molecule to another non-novel molecule, following which it undergoes minor changes over the next few steps to yield a novel molecule. Figure 3b shows the trajectory of a ChEMBL molecule with a 5% masking rate. In the first step, this molecule also changes from one training molecule to another non-novel molecule. Following this, further changes yield a novel molecule. The molecule evolves again in further iterations, albeit forming unexpected ring structures after 300 steps.

**Conditional generation.** In accordance with the framework proposed by Kwon et al.[25], we generate molecules conditioned on three different target values of the molecular weight (MolWt) and Wildman-Crippen partition coefficient (LogP) properties. We also compute KLD scores for the generated molecules. The KLD

**Table 5 Conditional generation results on QM9. Results for MGM are chosen from a range of sampling iterations and both initialization strategies.**

| Target Condition | Model | G-mean | Unique Count | Property Value | KLD Score |
|---|---|---|---|---|---|
| MolWt = 120 | NAT GraphVAE | 0.623 | 3048 | 124.47 ± 7.58 | 0.843 |
| | MGM | 0.522 | 8800 | 120.02 ± 7.66 | 0.811 |
| | MGM - Final Step | 0.404 | 8509 | 119.42 ± 7.67 | 0.761 |
| | Dataset | — | — | — | 0.679 |
| MolWt = 125 | NAT GraphVAE | 0.565 | 2326 | 127.21 ± 7.05 | 0.827 |
| | MGM | 0.561 | 9983 | 125.00 ± 8.48 | 0.850 |
| | MGM - Final Step | 0.354 | 9293 | 122.48 ± 7.20 | 0.936 |
| | Dataset | — | — | — | 0.835 |
| MolWt = 130 | NAT GraphVAE | 0.454 | 1204 | 129.12 ± 6.79 | 0.614 |
| | MGM | 0.501 | 9465 | 128.85 ± 8.85 | 0.705 |
| | MGM - Final Step | 0.369 | 8892 | 126.85 ± 7.43 | 0.789 |
| | Dataset | — | — | — | 0.695 |
| LogP = -0.4 | NAT GraphVAE | 0.601 | 2551 | −0.409 ± 0.775 | 0.739 |
| | MGM | 0.424 | 9506 | −0.349 ± 0.503 | 0.803 |
| | MGM - Final Step | 0.300 | 9495 | −0.337 ± 0.523 | 0.876 |
| | Dataset | — | — | — | 0.811 |
| LogP = 0.2 | NAT GraphVAE | 0.562 | 2188 | 0.051 ± 0.746 | 0.803 |
| | MGM | 0.378 | 9524 | 0.200 ± 0.468 | 0.846 |
| | MGM - Final Step | 0.376 | 9487 | 0.202 ± 0.462 | 0.895 |
| | Dataset | — | — | — | 0.816 |
| LogP = 0.8 | NAT GraphVAE | 0.515 | 1837 | 0.588 ± 0.759 | 0.807 |
| | MGM | 0.418 | 9360 | 0.769 ± 0.473 | 0.826 |
| | MGM - Final Step | 0.300 | 9294 | 0.745 ± 0.442 | 0.857 |
| | Dataset | — | — | — | 0.797 |

The results shown here correspond to the best mean property value (MGM) or the final sampling iteration with initialization chosen according to the better geometric mean among the five GuacaMol metrics (MGM—Final Step). Results for the NAT GraphVAE baseline model[25] that we trained are also shown. 'Dataset' rows refer to molecules sampled from the dataset with MolWt within ± 1 for the MolWt conditions and LogP within ± 0.1 for the LogP conditions. G-mean refers to the geometric mean of validity, uniqueness and novelty.

score is expected to decrease compared to unconditional generation since MolWt and LogP are two of the properties used to calculate this score; as these properties become skewed towards the target values, the similarity to the dataset will decrease. If a model maintains a reasonably high KLD score while achieving a mean property value close to the target value, it indicates that the other physiochemical properties of the generated molecules are similar to those of the dataset molecules. Conditional generation results for our model and the baseline Kwon et al.[25] model are shown in Table 5.

MGM generates molecules with property values close to the target value of the desired property. For the MolWt=120, MolWt=125, LogP=0.2 and LogP=0.8 conditions, the mean target property of the molecules generated by MGM is closer to the target value than of those generated by NAT GraphVAE. For the MolWt=130 and LogP=−0.4 conditions, the mean is slightly further. For LogP, MGM has lower standard deviations whereas for MolWt, NAT GraphVAE has slightly lower standard deviations. A lower standard deviation corresponds to more reliable generation of molecules with the target property. The G-means of validity, uniqueness and novelty are similar for both models on MolWt, and better for NAT GraphVAE on LogP.

The molecules generated by MGM have similar properties to the dataset molecules. This is reflected by the KL-divergence scores, which are generally higher for MGM than for NAT GraphVAE and greater than 0.8 in all cases but one. The KLD scores in the Dataset rows of Table 5 are considerably less than 1, showing the decrease in similarity to the full dataset as the MolWt or LogP values are skewed. MGM achieves a higher KL-divergence score than Dataset in the majority of cases. This indicates that MGM produces molecules that are optimized for the target property while maintaining physiochemical similarity to the dataset distribution.

**Table 6 Time taken for training and generation.**

| Dataset | Training time per epoch (min) | Generation time/sample/sampling iteration (sec) |
|---|---|---|
| QM9 | 6 | 0.00542 |
| ChEMBL | 280 | 0.00622 |

Generation time/sample/sampling iteration is measured as: $\frac{\text{time taken to carry out 100 sampling iterations for a batch of } J \text{ samples}}{100J}$ For QM9, $J = 2500$ whereas for ChEMBL, $J = 1500$ due to memory constraints.

The results for MGM—Final Step approach slightly differ from those for MGM. By design, the mean values of the target properties are a little further from the target values than for MGM. Compared with MGM, the standard deviations and G-means for MGM—Final Step are generally lower while the KL-divergence scores are higher.

**Computational efficiency**. Time taken to train and generate from models is shown in Table 6. For each sample, generation time per sampling iteration is low (on the order of milliseconds), as the forward pass through the neural network is computationally cheap and many molecules can be processed in parallel. The ChEMBL model takes longer than the QM9 model for generation as it has more MPNN layers and also because ChEMBL molecules are on average larger than QM9 molecules. The ChEMBL model takes longer to train per epoch than the QM9 model for the same reasons and also because ChEMBL has many more molecules than QM9. Note that training time for ChEMBL could be significantly lowered if dynamic batching strategies are used so that the batch size is not constrained by

the size of the largest molecule in the dataset. See the Datasets and Evaluation part of the Methods section for more details on the datasets. We use one Nvidia Tesla P100-SXM2 GPU with 16 GB of memory for all our experiments; the use of multiple GPUs or a GPU with larger memory would further increase computational speed.

## Discussion

In this work, we propose a masked graph model for molecular graphs. We show that we can sample novel molecular graphs from this model by iterative sampling of subsets of graph components. Our proposed approach models the conditional distributions of subsets of graph components given the rest of the graph, avoiding many of the drawbacks of previously proposed models, such as expensive marginalization and fixing an ordering of variables.

We evaluate our approach on the GuacaMol distribution-learning benchmark on the QM9 and ChEMBL datasets. We find that the benchmark metrics are correlated with each other, so models and generation configurations with higher validity, KL-divergence and Fréchet ChemNet Distance scores usually have lower novelty scores. Hence evaluating models based on the trade-off between different metrics may be more informative than evaluating them based on a heuristic such as the sum of the metrics. We observe that by varying generation hyperparameters, our model balances these metrics more efficiently than previous state-of-the-art baseline models.

For some applications, it is convenient to evaluate results based on one masking rate rather than evaluating this trade-off. A discussion of how to choose this generation hyperparameter is given under the Model Architecture, Training and Unconditional Generation Details part of the Methods section. We recommend using a generation masking rate corresponding to masking out 5-10 edges of a complete graph having the median number of nodes in the dataset.

We show that on distribution-learning metrics, overall our model outperforms baseline graph-based methods. We also observe that our model is comparable to SMILES-based approaches on both datasets, but underperforms the LSTM, Transformer Small and Transformer Regular SMILES-based autoregressive models on ChEMBL. There are several differences between the QM9 and ChEMBL datasets (see the Datasets and Evaluation part of the Methods section) that could account for this, including number of molecules, median molecule size and presence of chirality information. There has also been extensive work in developing language models compared to graph neural networks, which may account for the greater success of the LSTM and Transformers. Furthermore, the ChEMBL dataset is provided as SMILES strings and the GuacaMol benchmark requires that graph representations be converted into SMILES strings before evaluation. This may advantage approaches that work with SMILES strings directly rather than converting to and from graph representations of molecules. Although there are molecular benchmarks for evaluating different aspects of machine learning-based molecular generation[46,54], they use string representations of molecules and do not evaluate graph-level properties. Developing datasets and benchmarks that incorporate graph-level information that is not readily encoded as strings, such as spatial information, would alleviate this issue. We leave further investigation into the reasons behind the difference in performance to future work.

From our observations of molecular trajectories, we see that molecules converge towards the space of dataset molecules regardless of whether training or marginal initialization is used.

This verifies that the sampler produces molecules from the distribution that it was trained on. We also see that using a higher masking rate results in greater changes between sampling iterations and molecules that are less similar to the dataset used. We hypothesize that this is the case for two reasons. First, a greater proportion of the graph is updated at each step. Second, the predictive distributions are formed from a graph with a greater proportion of masked components, resulting in higher entropy.

We carry out conditional generation, observing that our model captures the target properties of molecules better than a baseline graph-based generative model while maintaining similarity of the generated molecules to the distribution of dataset molecules.

Finally, we observe the computational cost of our models and note that generation time per molecule is low after training the model.

Future avenues of work include incorporating additional information such as inter-atomic distances into our graph representations. In the GuacaMol benchmark[46], for example, the data is provided as strings and must be converted back into strings for evaluation. Hence features that are not readily encoded as strings are not used by either the text-based or graph-based models, and cannot be a part of evaluation. The development of benchmarks that account for the spatial nature of molecules, for example by incorporating 3D coordinates, would help highlight the advantages of graph-based generative models compared to SMILES-based models.

As discussed in the Model Architecture, Training and Unconditional Generation Details part of the Methods section, using the same masking rate for molecules of different sizes results in a disproportionately large number of 'prospective' edges being masked out for large molecules, which is problematic for our MPNN to handle. Finding a way to address this problem would be beneficial in scaling this work to larger molecules.

Another direction is to make our model semi-supervised. This would allow us to work with target properties for which the ground-truth cannot be easily calculated at test time and only a few training examples are labelled. Our work can also be extended to proteins, with amino acids as nodes and a contact map as an adjacency matrix. Conditional generation could be used in this framework to redesign proteins to fulfil desired functions. Furthermore, although we use the principle of denoising a corrupted graph for learning the joint distribution, the same procedure could be adapted for lead optimization. Finally, as our approach is broadly applicable to generic graph structures, we leave its application to non-molecular datasets to future work.

## Methods

**Model architecture**. A diagram of our model including featurization details is given in Fig. 4. We start by embedding the vertices and edges in the graph $G_{\setminus\eta}$ to get continuous representations $\mathbf{h}_{v_i} \in \mathbb{R}^{d_0}$ and $\mathbf{h}_{e_{ij}} \in \mathbb{R}^{d_0}$ respectively, where $d_0$ is the dimensionality of the continuous representation space[55]. We then pass these representations to a message passing neural network (MPNN)[56]. We use an MPNN as the fundamental component of our model because of its invariance to graph isomorphism. An MPNN layer consists of an aggregation step that aggregates messages from each node's neighboring nodes, followed by an update step that uses the aggregated messages to update each node's representation. We stack $L$ layers on top of each other to build an MPNN; parameters are tied across all $L$ layers. For all except the last layer, the updated node and edge representations output from layer $l$ are fed into layer $l + 1$. Unlike the original version of the MPNN, we also maintain and update each edge's representation at each layer. Any variant of a graph neural network that effectively models the relationships between node and edge features can be used, such as an MPNN. Our specific design is described below.

Diagrams of our MPNN's node and edge update steps are given in Supplementary Figure 5. At each layer $l$ of the MPNN, we first update the hidden state of each node $v_i$ by computing its accumulated message $\mathbf{u}_{v_i}^{(l)}$ using an

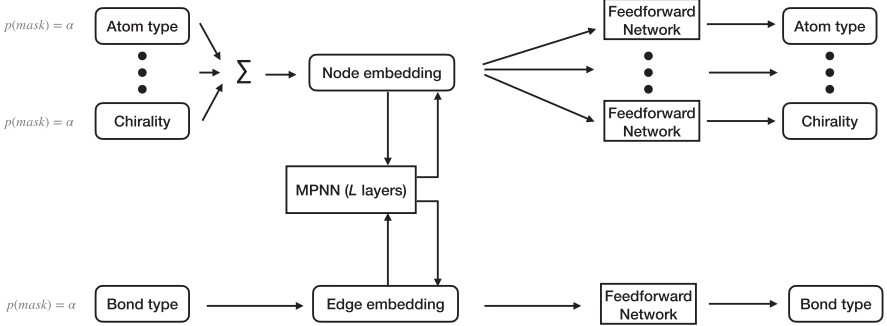

**Fig. 4 Model architecture.** A description of the node and edge features is given in the Property Embeddings part of the Methods section.

aggregation function $J_\nu$ and a spatial residual connection $R$ between neighboring nodes:

$$\mathbf{u}_{v_i}^{(l)} = J_\nu \left( \mathbf{h}_{v_i}^{(l-1)}, \left\{ \mathbf{h}_{v_j}^{(l-1)} \right\}_{j \in N(i)}, \left\{ \mathbf{h}_{e_{i,j}}^{(l-1)} \right\}_{j \in N(i)} \right) + R \left( \left\{ \mathbf{h}_{v_j}^{(l-1)} \right\}_{j \in N(i)} \right), \quad (4)$$

$$J_\nu \left( \mathbf{h}_{v_i}^{(l-1)}, \left\{ \mathbf{h}_{v_j}^{(l-1)} \right\}_{j \in N(i)}, \left\{ \mathbf{h}_{e_{i,j}}^{(l-1)} \right\}_{j \in N(i)} \right) = \sum_{j \in N(i)} \mathbf{h}_{e_{i,j}}^{(l-1)} \cdot \mathbf{h}_{v_j}^{(l-1)}, \quad (5)$$

$$R \left( \left\{ \mathbf{h}_{v_j}^{(l-1)} \right\}_{j \in N(i)} \right) = \sum_{j \in N(i)} \mathbf{h}_{v_j}^{(l-1)}, \quad (6)$$

$$\mathbf{h}_{v_i}^{(l)} = \text{LayerNorm} \left( \text{GRU} \left( \mathbf{h}_{v_i}^{(l-1)}, \mathbf{u}_{v_i}^{(l)} \right) \right), \quad (7)$$

where $N(i)$ is the set of indices corresponding to nodes that are in the one-hop neighbourhood of node $v_i$. GRU[53] refers to a gated recurrent unit which updates the representation of each node using its previous representation and accumulated message. LayerNorm[57] refers to layer normalization.

Similarly, the hidden state of each edge $\mathbf{h}_{e_{i,j}}$ is updated using the following rule for all $j \in N(i)$:

$$\mathbf{h}_{e_{i,j}}^{(l)} = J_e \left( \mathbf{h}_{v_i}^{(l-1)} + \mathbf{h}_{v_j}^{(l-1)} \right). \quad (8)$$

The sum of the two hidden representations of the nodes incidental to the edge is passed through $J_e$, a two-layer fully connected network with ReLU activation between the two layers[58,59], to yield a new hidden edge representation.

The node and edge representations from the final layer are then processed by a node projection layer $A_\nu : \mathbb{R}^{d_0} \to \Lambda^T$ and an edge projection layer $A_e : \mathbb{R}^{d_0} \to \Lambda^R$, where $\Lambda^T$ and $\Lambda^R$ are probability simplices over node and edge types respectively. The result is the distributions $p(v|G_{\backslash\eta})$ and $p(e|G_{\backslash\eta})$ for all $v \in \mathcal{V}$ and all $e \in \mathcal{E}$.

### Property embeddings

*Node property embeddings.* We represent each node using six node properties indexed as $\{\kappa \in \mathbb{Z} : 1 \le \kappa \le 6\}$, each with its own one-hot embedding. The properties are obtained using RDKit[60]. Each node in a graph corresponds to a heavy atom in a molecule. During the forward pass, each of these embeddings is multiplied by a separate weight matrix $W_\kappa \in \mathbb{R}^{T_\kappa \times d_0}$, where $T_\kappa$ is the number of categories for property $\kappa$. The resulting continuous embeddings are summed together to form an overall embedding for the node. The entries of the one-hot embeddings for each of the properties are:

- Atom type: chemical symbol (e.g. C, N, O) of the atom;
- Number of hydrogens: number of hydrogen atoms bonded to the atom;
- Charge: net charge on the atom, where the first index represents the minimum charge on an atom in the dataset and the last index represents the maximum;
- Chirality type: unspecified, tetrahedral clockwise, tetrahedral counter-clockwise, other;
- Is-in-ring: atom is or is not part of a ring structure;
- Is-aromatic: atom is or is not part of an aromatic ring.

Each one-hot embedding also has an additional entry corresponding to the MASK symbol.

After processing the graph with the MPNN, we pass the representation of each node through six separate fully-connected two-layer networks with ReLU activation between the layers. For each node, the output of each network is a distribution over the categories of the initial one-hot vector for one of the properties. During training, we calculate the cross-entropy loss between the predicted distribution and the ground-truth for all properties that were masked out by the corruption process.

The choice of nodes for which a particular property is masked out is independent of the choice made for all other properties. The motivation for this is to allow the model to more easily learn relationships between different property types. The atom-level property information that we use in our model is the same as that provided in the SMILES string representation of a molecule. We also tried masking out all features for randomly selected nodes, but this yielded a significantly higher cross-entropy loss driven largely by the atom type and hydrogen terms.

Since the ChEMBL dataset does not contain chirality information, the chirality type embedding is superfluous for ChEMBL.

We note from preliminary experiments that using fewer node features, specifically only the atom type and number of hydrogens, results in a substantially higher cross-entropy loss than using all the node features listed above.

*Edge property embeddings.* We use the same framework as described for node property embeddings. We only use one edge property with the weight matrix $\mathcal{W} \in \mathbb{R}^{R \times d_0}$, whose one-hot embedding is defined as follows:

- Bond type: no, single, double, triple or aromatic bond.

### Model architecture, training and unconditional generation details.

For the QM9 dataset, we use one 4-layer MPNN, with parameter sharing between layers. For the ChEMBL dataset, we use one 6-layer MPNN with parameter sharing. We experiment with using more layers for ChEMBL in case more message passing iterations are needed to cover a larger graph. The results of an extensive hyperparameter search on ChEMBL are given in Supplementary Table 2. For both datasets, we use an embedding dimensionality $d_0 = 2048$. We use the Adam optimizer[61] with learning rate set to 0.0001, $\beta_1 = 0.9$ and $\beta_2 = 0.98$. We use a batch size of 800 molecules for QM9 and 512 molecules for ChEMBL. For ChEMBL, we perform 16 forward-backward steps with minibatches of 32 each to compute the gradient of the minibatch of 512 molecules, in order to cope with the limited memory size on a GPU. We clip the gradient for its norm to be at most 10.

During training, we uniformly at random mask each node feature (including atom type) and edge feature (including bond type) with probability $\alpha$, while randomly varying $\alpha$ uniformly between 0 and 0.2. Nodes are considered as neighbors in the MPNN if they are connected by an edge that is either masked out, or does not have bond type no-bond. For the purposes of masking, the total number of edges in the graph is $\frac{|V|(|V|-1)}{2}$ i.e. every possible node pair (excluding self-loops) in the symmetric graph is considered as a 'prospective edge' that can be masked out. During validation, we follow the same procedure but with $\alpha$ fixed at 0.1, so that we can clearly compare model checkpoints and choose the checkpoint with the lowest validation loss for generation.

For QM9, we carry out generation experiments while using a masking rate of either 10% or 20%, corresponding to the mean and maximum masking rates during training respectively. For ChEMBL, we use a masking rate of either 1% or 5%, as we found that the higher masking rates led to low validity scores in our preliminary experiments. The number of prospective edges masked and replaced for a median ChEMBL molecule with a 1% masking rate and for a median QM9 molecule with a 10% masking rate are both approximately 4. This indicates that the absolute number rather than portion of components masked out directly impacts generation quality. For a constant masking rate, the number of masked out prospective edges scales as the square of the number of nodes in the graph. The number of bonds in a molecule does not scale in this way; larger molecules are likely to have sparser adjacency matrices than small molecules. Masking out a very large number of prospective edges could degrade performance as this would yield an unnaturally dense graph to the MPNN. This is because every prospective edge of type 'no edge' that is masked out would appear as an edge to the MPNN. This would result in message passing between many nodes in the input graph that are far apart in the sparse molecule. We therefore propose a masking rate corresponding to masking out a similar number of prospective edges (approximately 5–10) when using MGM on other datasets. Nevertheless, finding an automated way of setting the masking rate would be a valuable direction for future research.

We use the same independence constraint during generation as we use during training when choosing which properties to mask out for each node or edge. We vary the initialization strategy between training and marginal initialization.

For QM9, we run 400 sampling iterations sequentially to generate a sequence of sampled graphs. For ChEMBL, we run 300 iterations. We calculate the GuacaMol evaluation metrics for our samples after every generation step for the first 10 steps, and then every 10–20 steps, in order to observe how generation quality changes with the number of generation steps.

**Conditional generation details.** We carry out conditional generation corresponding to two different molecular properties: molecular weight (MolWt) and the Wildman–Crippen partition coefficient (LogP). We train a separate model for each property on QM9, with the same hyperparameters as used for the unconditional case. For each property, we first normalize the property values by subtracting the mean and dividing by the standard deviation across the training data. We obtain an embedding of dimension $d_0$ for the property by passing the one-dimensional standardized property value through a two-layer fully-connected network with ReLU activation between the two layers. We add this embedding to each node embedding and then proceed with the forward pass as in the unconditional case. For generation, we use a 10% masking rate and carry out 400 sampling iterations with both training and marginal initializations.

We evaluate 10,000 generated molecules using the framework outlined by Kwon et al.[25] in their work on non-autoregressive graph generation. This involves computing summary statistics of the generated molecules for target property values of 120, 125 and 130 for MolWt, and −0.4, 0.2 and 0.8 for LogP. We choose results corresponding to the initialization and number of sampling iterations that yield the mean property value that is closest to the target value. We also provide results from the final generation step with the initialization corresponding to the higher geometric mean among the five GuacaMol metrics.

Finally, we calculate KLD scores for molecules from the QM9 dataset with property values close to the target values. For the MolWt conditions, we sample 10,000 molecules from the dataset that have a MolWt within 1 of the target MolWt. For the LogP conditions, we sample 10,000 molecules from the dataset that have LogP value within 0.1 of the target LogP value.

**Details of baseline models.** We train two variants of the Transformer[3] architecture: Small and Regular. The Transformer Regular architecture consists of 6 layers, 8 attention heads, embedding size of 1024, hidden dimension of 1024, and dropout of 0.1. The Transformer Small architecture consists of 4 layers, 8 attention heads, embedding size of 512, hidden dimension of 512, and dropout of 0.1. Both Transformer-Small and -Regular are trained with a batch size of 128 until the validation cross-entropy loss stops improving. We set the learning rate of the Adam optimizer to 0.0001, $\beta_1 = 0.9$ and $\beta_2 = 0.98$. The learning rate is decayed based on the inverse square root of the number of updates. We use the same hyperparameters for the Transformer Small and Regular models on both QM9 and ChEMBL.

We follow the open-source implementation of the GuacaMol benchmark baselines at https://github.com/BenevolentAI/guacamol_baselines for training an LSTM model on QM9. Specifically, we train the LSTM with 3 layers of hidden size 1024, dropout of 0.2 and batch size of 64, using the Adam optimizer with learning rate 0.001, $\beta_1 = 0.9$ and $\beta_2 = 0.999$. We do not train the rest of the baseline models ourselves. For QM9: CharacterVAE[49], GrammarVAE[50], GraphVAE[23], and MolGAN[51] results are taken from Cao and Kipf[51]. For ChEMBL: AAE[6], ORGAN[62], Graph MCTS[52], VAE, and LSTM results are taken from Brown et al.[46]. NAT GraphVAE results are taken from Kwon et al.[25] for ChEMBL. To carry out unconditional and conditional generation from NAT GraphVAE on QM9, we train a model using the publicly available codebase provided by the paper's authors at https://github.com/seokhokang/graphvae_approx.

**Datasets and evaluation.** We evaluate our approach using two widely used[23,49,63] datasets of small molecules: QM9[43,44], and a subset of the ChEMBL database[45] (Version 24) as defined by Fiscato et al.[64] and used by Brown et al.[46]. All references to ChEMBL in this paper are references to this subset of the database. Heavy atoms and bonds in a molecule correspond to nodes and edges in a graph, respectively.

The QM9 dataset consists of approximately 132,000 molecules with a median and maximum of 9 heavy atoms each. Each atom is of one of the following $T = 4$ types: C, N, O, and F. Each bond is either a no-bond, single, double, triple or aromatic bond ($R = 5$). The ChEMBL dataset contains approximately 1,591,000 molecules with a median of 27 and a maximum of 88 heavy atoms each. It contains 12 types of atoms ($T = 12$): B, C, N, O, F, Si, P, S, Cl, Se, Br, and I. Each bond is either a no-bond, single, double, triple or aromatic bond ($R = 5$).

The QM9 dataset is split into training and validation sets, while the ChEMBL dataset is split into training, validation and test sets. We use the term dataset distribution to refer to the distribution of the combined training and validation sets for QM9, and the combined training, validation and test sets for ChEMBL. Similarly, we use the term dataset molecule to refer to a molecule from the combined QM9 or ChEMBL dataset.

To numerically evaluate our approach, we use the GuacaMol benchmark[46], a suite of benchmarks for evaluating molecular graph generation approaches. The GuacaMol framework operates on SMILES strings, so we convert our generated graphs to SMILES strings before evaluation. Specifically, we evaluate our model using distribution-learning metrics from GuacaMol: the validity, uniqueness, novelty, KL-divergence[47] and Fréchet ChemNet Distance[48] scores. GuacaMol uses 10,000 randomly sampled molecules to calculate each of these scores. Validity measures the ratio of valid molecules, uniqueness estimates the proportion of generated molecules that remain after removing duplicates and novelty measures the proportion of generated molecules that are not dataset molecules. The KL-divergence score compares the distributions of a variety of physiochemical descriptors estimated from the dataset and a set of generated molecules. The Fréchet ChemNet Distance score[48] measures the proximity of the distribution of generated molecules to the distribution of the dataset molecules. This proximity is measured according to the Fréchet Distance in the hidden representation space of ChemNet, which is trained to predict the chemical properties of small molecules[65].

## Data availability

The datasets used in this work are publicly available. They are referenced in the Datasets and Evaluation part of the Methods section.

## Code availability

Code, pretrained MGM models, training and generation scripts for MGM and baseline models, and lists of generated molecules can be found at https://github.com/nyu-dl/dl4chem-mgm[66].

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

## Acknowledgements
O.M. would like to acknowledge NRT-HDR: FUTURE. K.C. thanks Naver, eBay and NVIDIA.

## Author contributions
O.M., E.M. and K.C. conceived the initial idea and started the project. O.M. wrote the code and ran most of the experiments with the help of E.M. O.M., E.M., R.B. and K.C. wrote the paper and conceived improvements to the experiments. EM worked on this project during his time at New York University.

## Competing interests
The authors declare no competing interests.
