## [Peer Review File · Nature Communications]

REVIEWER COMMENTS

Reviewer #1 (Remarks to the Author):

This paper introduces a model for generating a novel molecular graph type structure that satisfies the target properties to adapt to various evaluation components for generating new molecules. In particular, the proposed model was evaluated not only for validity, novelty, and uniqueness, but also for whether the model was reliably trained with KL-divergence. In addition, the model is evaluated for its similarity to a real-world molecular structure using Frechet ChemNet Distance Score (FCD).

In my view this work could have great impact in the field of generating a novel molecular structures with machine learning approaches, however the performance and novelty of the model making the compounds is not very clear. The authors present a comprehensive comparison of their model to similar models in performance, but the algorithm proposed in Table 3 and Table 4 is not clear about the improved performances compared to other methods. Specifically, in Table 4, the LSTM model seems to be the most performance in terms of comprehensive scores over MGM models.

Question:

- 1) Please define the limitations of previous models and describe numerically how the MGM solved them.
- 2) Please explain in detail what part of the previous model you modified or proposed to improve the FCD performance.
- 3) I think it will be improved if we train the previous models (ex. NAT GraphVAE) with the FCD value as the target properties, what do you think?

Reviewer #2 (Remarks to the Author):

Summary

First of all, I would like to give kudos to the authors on an extremely well-written paper. Many of the questions that I had as I was reading the paper were eventually answered in different areas of the Methods or SI, which I thought was very well done so that it does not disrupt the flow of the "main" text. Overall it was enjoyable to read this paper. Explanations were generally very clear (although I still have some questions which I outline below). I was also very happy to see that the code was provided and that it appeared to be clear and readable code. From a reproducibility standpoint, I think this manuscript would score very high as the authors did a good job of reporting what was done plus have made their code available.

Now to the science. In this work, the authors present a graph-based molecular generative model that operates by masking different elements on input graphs uniformly at random, learning the conditional distributions for the masked graphs, and then generating new graphs using their model. They refer to their approach as masked graph modeling. I believe the model is very interesting, although not perhaps so "powerful" as to say it competes with SOTA deep molecular generative models, given that the model can only learn to complete nearly complete graphs (correct me if I misunderstood). As such, it appears that the model would be quite interesting from a lead optimization point of view (if thinking about it from the context of drug discovery), but if, let's say, the goal of the model was to find drug candidates with a completely different scaffold (e.g. more analogous to hit discovery) the model would be unable to do so (again, please correct me if I am wrong). As such, I think the authors could improve the impact of their method by "branding" it more as a tool for molecular optimization rather than as an outright de novo design tool.

As a deep molecular generative model, the model is ok. I am not impressed by the performance which in my opinion is very average, and I do not see this being implemented in e.g. a drug discovery pipeline, but I still find the approach very interesting. The authors also do a very

interesting analysis. I have divided my feedback below into Major and Minor points.

Major Points

1) For starters, I thought it was a shame that the performance of the models significantly drops when masking more than 20% of the nodes, as my understanding is that being able to mask more nodes would possibly improve the diversity of the structures generated. Is this just a limitation of the underlying model used, or do you believe that greater masking percentages could be used by tuning the model (somehow)? Just curious.

2) To clarify, in neither generation approach (i.e. using training or marginal initialization) can the model generate graphs starting from an empty graph, is this correct? No matter how the graphs are initialized for generation, they will have at most the unmasked % number of graph elements (nodes + edges). That is, if the mean # nodes in training set graphs is 30, and a 10% masking rate was used, then the graphs will be initialized with ~27 nodes? Looking for some clarification as this was the part that was not entirely clear in the text for me.

3) I was wondering why a masking rate of 1-5% was used for ChEMBL and 10-20% for QM9? That is, it doesn't make sense to me that the performance of MGM would decrease faster with decreasing masking rate on ChEMBL, which has the larger structures, than on QM9. I think the authors tried to explain this in the text but it was not entirely clear the explanation.

4) As the model is carrying out iterative graph generation, it is not clear to me how/when the model knows to "stop" the graph generation? Similarly, in the graph generation process, how do you set how many nodes/edges the graph can have (i.e. the max value)? Can your model generate graphs larger than are in the training set? My understanding is not, as the model would not have seen anything like that before (should you initialize a larger graph than was seen in the training set), but again just looking for some clarification. A schematic for the graph generation process could be useful here.

5) Which initialization method is used for the results shown in Table 1 & Figure 1 (training or marginal)? Also, would they lead to different results?

6) (this is related to point 2 above) I don't fully understand what 1% masking corresponds to in the models. To illustrate, I believe most molecular graphs in ChEMBL have between 20-50 heavy atoms (and let's say roughly 2 edges per node), which if we assume an average of 40 nodes per structures, means there are ~120 elements (nodes and or edges) in each graph to mask. So at 1% masking, does it mean each graph in the training set only has ~1.2 nodes or ~1.2 edges masked? On a related note, on pg. 14, the authors state "We recommend using a generation masking rate corresponding to masking out 5-10 edges of a complete graph having the median number of nodes in the dataset." but I don't see how 5-10 edges corresponds to 1-5% masking rate in ChEMBL and 10-20% in QM9. And once again, on pg. 17, the authors say "The number of edges masked and replaced for a median ChEMBL molecule with a 1% masking rate and for a median QM9 molecule with a 10% masking rate are both approximately 4." How does a 1% masking rate in ChEMBL corresponds to 4 edges, as there are not 400 edges on the typical ChEMBL molecule, but much fewer... Am I misunderstanding something? It could help to illustrate the masking process by including a scheme (maybe in the SI), though this is just a suggestion.

7) In Table 4, the authors compare to other SOTA models, and say their "masked graph model corresponds to 1% masking rate and training graph initialization". If I understand correctly, then this means the authors start generation from a nearly complete graph and add the remaining 1% of nodes/edges. I am guessing that, in comparison, the LSTM is starting from an "empty" molecular string, adding tokens to the string until sampling the termination token. So, a more "fair" comparison for MGM against the LSTM would be to start from the SMILES corresponding to the "masked" training set structure and letting the model complete them, no? I suppose the autoencoder and GAN models used in Figure 4 generate new structures in a "one-shot" manner. For the Graph MCTS, I believe it also starts from molecular fragments. So comparing to the GAN/VAE/MCTS methods makes "sense", while comparison to the LSTM is a bit like comparing apples and oranges. Can you comment a bit on this? I am not suggesting to remove comparison to the LSTM, as the LSTM has SOTA performance for some generation tasks and this is obviously an interesting comparison, but just to discuss a bit more what is being compared.

8) I would recommend part of the background in Appendix A to be discussed in the main text, namely a bit more background on graph-based generative models. I say this because on pg. 9 (and other places throughout the manuscript), the authors claim that their approach "outperforms" existing graph-based methods, which I would say is a bit of an overstatement, given that the

model was only compared to “old” graph-based generative models. While models like MolGAN are definitely good “baselines”, it is definitely not SOTA. So I believe it is important to mention that there are plenty of other graph-based generative models which have been since published (e.g. DEFactor by Assouel et al (2018), MolGAN by De Cao et al (2018), RVAE by Ma et al (2018), GraphVAE by Simonovsky et al (2018), GraphNVP by Madhawa et al (2019), MoFlow by Zang et al (2020), NeVAE by Samanta et al (2018), MolRNN by Li et al (2018), JTN-VAE by Jin et al (2018), some models by DeepMind, Li et al (2018), HierVAE by Jin et al (2020), GraphINVENT by Mercado et al (2020), GCPN by You et al (2018), GraphAF by Shi et al (2020), MolecularRNN by Popova et al (2019), to give just a few examples; I saw after I wrote this that many of these were already discussed in the SI, which was great, maybe the authors could pick a few of the better performing graph-based generative models to discuss in the main text as well). Given that this is a publication on a graph-based molecular generative model, I think it is appropriate that the authors expand a bit here, and provide more background to the reader.

9) Curious how the model performance changes when fewer/additional atom features are included (if this was explored at all)?

10) Also curious how long a typical training run takes for MGM? What about a generation run? Let’s say, for the QM9 dataset and on whichever GPU you used. I’m just trying to get an idea for what the computational cost is for the model, because I assume it is relatively “cheap” for a graph-based generative model - and if this is the case I believe you can (and should) “sell” this aspect more in your manuscript. If you have numbers for the specific training time per epoch that would be interesting to see, too.

11) On pg. 14, the authors say “Developing datasets and benchmarks that incorporate graph-level information that is not readily encoded as strings, such as spatial information, would alleviate this issue.” yet there do exist other benchmarks (e.g. molecular rediscovery benchmarks, which are actually implemented in GuacaMol and it is unclear why the authors did not use them; also benchmarks related to the coverage of chemical space, see Arús-Pous et al (2019) <https://jcheminf.biomedcentral.com/articles/10.1186/s13321-019-0393-0>). So this statement should be reworded, as other benchmarks do exist, so are these benchmarks not sufficient to address the string vs graph comparison (and if so, why not?), or are the authors planning on developing another benchmark anyhow?

12) On pg. 14, there is this statement: “Future avenues of work include incorporating additional information such as inter-atomic distances into our graph representations. The development of benchmarks that account for the graphical representations of molecules, for example by incorporating spatial information, would help more fairly compare graph-based generative models to each other and to SMILES based models.” I disagree with this statement, actually. Since the string-based representation is built on the graph-representation, I believe the string- and graph-based generative models (for 2D molecules) are entirely comparable if they have used the same node- and edge-features in their respective representations. So I would recommend to the authors to rephrase this statement (or if not, let me know why you disagree with me). Nonetheless, I do think generation of 3D structures from either string- or graph-based representations is extremely interesting!

13) On pg. 17, the authors say “We use more layers for ChEMBL because more message passing iterations are needed to cover a larger graph.” Was this determined through some kind of hyperparameter optimization procedure or is this just a “guess”? From what I have seen in the molecular property prediction and molecular generation literature, the number of “optimal” message passes in an MPNN does not necessarily correlate with the size of the graphs in the training set. One could also make the argument that if the model is searching for certain “pharmacophores” (e.g. in toxicity prediction), even if training set graphs are large the optimal number of message passes might still be small for toxicity prediction. So what I am saying is that I would like more details on how the authors determined the optimal number of message passes, even if they just chose these numbers based on intuition.

Minor Points

14) In Figure 1, why is the correlation not as smooth for the QM9 dataset as when the model was trained on ChEMBL? That is, why the bump in the MGM curves (for both masking %) in Figure 1a? Maybe it is hard to know why, but just curious if you had an idea.

15) I like Table 1, showing the correlation between different Guacamol metrics. Based on these correlations, the authors suggest that they can look at a subset of the metrics, namely the

uniqueness, Fréchet and novelty scores, to gauge generation quality. I thought this was really valuable.

16) I also like the idea that the smoother change in the Fréchet ChemNet Distance score vs the novelty allows the authors a greater degree of control in MGM compared to other molecular generative models. I found this an interesting metric and a cool angle from which to look at molecular generative model performance.

17) Were heavy atoms only considered as nodes, or hydrogens as well? (the list on pg. 16 makes me think only heavy atoms but just wanted to confirm)

18) Some of the molecules in Figure 3 are just crazy (from a chemistry standpoint). My recommendation is to pick another example to show for e.g. Figure 3a (esp after 400 steps). I personally don't care what example is shown (and I actually like this example, I find it very useful from the perspective of deep generative models), but since this is not a dedicated computational journal, I think maybe it is worth updating the example you show. If a chemist were to see these "examples", my guess is they would not be inclined to use your model. Of course, your goal may not be to convince a chemist to use your model per se (at least not at the moment), so this is highly subjective advice...

19) In pg. 16, how were these properties computed for the molecules in the training set, from looking at the code I figured RDKit but will be good to mention it in the paper too.

20) On pg. 14, the authors say "There are several differences between the QM9 and ChEMBL datasets that could account for this, including number of molecules, median molecule size and presence of chirality information." Here you could point the reader to the section in the Methods, since there you describe what the specific differences are.

21) Just a suggestion, but for the GitHub repo, I recommend to include an environment file so users can easily install the *exact* same dependencies on their system as you have used. Even sometimes installing packages in a slightly different order leads to slightly different dependencies, and I have seen this lead to problems (especially when PyTorch, tensorflow, and RDKit are involved).

Reviewer #3 (Remarks to the Author):

The generation of novel and sensible molecular structures is a key problem in computer aided drug discovery. The authors describe a novel approach for molecule generation based on masked graph models (MGM), which is complementary to previous approaches based on string representations or graph representations. Furthermore, results on the conditional generation of molecules are presented, where molecules with desirable molecular property ranges are generated.

Furthermore the authors perform a critical analysis of the recently proposed guacamol metrics, showing that some are correlated, and perform a multi-criterial assessment of the models to highlight that MGM can be tuned well to populate the full pareto front, whereas SMILES-based models are not as easy to control.

The paper also has interesting results on SMILES transformers, which underperform SMILES-LSTM models, which contrasts recent results in machine translation and NLP.

Overall this is a valuable contribution, which I would recommend for publication after the following points have been addressed.

Questions:

Is it really a surprise that novelty and fréchet distance (FCD) are negatively correlated? After all, the FCD measures the distance between the sampled molecules to a set of known molecules, whereas novelty measures how

Note that in the context of molecular design novelty by itself is of limited value, since we can trivially generate new molecules with unnatural structures.

Missing citations:

It would be good cite the original paper which introduced autoregressive LM type models for molecules (SMILES-LSTM) <https://arxiv.org/abs/1701.01329>

Additionally, I would recommend to cite work on reaction-driven molecule design, where molecules are generated by performing virtual chemical reactions, e.g. <https://arxiv.org/abs/2012.11522>

Reviewer #1 (Remarks to the Author):

This paper introduces a model for generating a novel molecular graph type structure that satisfies the target properties to adapt to various evaluation components for generating new molecules. In particular, the proposed model was evaluated not only for validity, novelty, and uniqueness, but also for whether the model was reliably trained with KL-divergence. In addition, the model is evaluated for its similarity to a real-world molecular structure using Frechet ChemNet Distance Score (FCD).

In my view this work could have great impact in the field of generating a novel molecular structures with machine learning approaches, however the performance and novelty of the model making the compounds is not very clear. The authors present a comprehensive comparison of their model to similar models in performance, but the algorithm proposed in Table 3 and Table 4 is not clear about the improved performances compared to other methods. Specifically, in Table 4, the LSTM model seems to be the most performance in terms of comprehensive scores over MGM models.

We thank the reviewer for their constructive comments. Our approach to generating molecular structures is novel; our aim is to show that this type of approach is successful in addition to the popular autoregressive generation approaches used by most deep learning models. Our model achieves higher scores across most metrics compared to existing graph-based models, and is competitive with the best SMILES-based models. For example, on ChEMBL our approach yields molecules that are similar to the training distribution as measured by FCD score, whereas the graph-based models achieve ~0 FCD score. We have also further clarified in the text that a) care must be taken when using the term outperform given that the evaluation metrics are correlated b) our model provides greater controllability over the evaluation metrics than existing approaches. We hope that additional graph-based benchmarks will be developed in the future to further evaluate non-autoregressive and autoregressive molecular generation models.

Question:

1) Please define the limitations of previous models and describe numerically how the MGM solved them.

Most previous models use either an autoregressive or latent variable based method for molecule generation. Each of these makes certain limiting assumptions.

Using an autoregressive model necessitates choosing an arbitrary ordering of nodes and edges in the molecule. On the other hand, with MGM there is no need to choose an ordering.

Using a latent variable model necessitates carrying out approximate rather than exact inference and/or carrying out expensive marginalisation over latent variables. On the other hand, our approach circumvents these issues entirely.

We have added clarifying statements in the text (page 4) to highlight that the methods we tested exhibit poor control over the tradeoff between different evaluation metrics. In comparison, MGM provides a natural way to control this tradeoff via the initialisation, number of sampling iterations and masking rate as shown in Figure 1 and Supplementary Figure 2. In addition, MGM outperforms the baseline graph based generative models that we test with a simple maximum

likelihood objective, without the need for reinforcement learning objectives as for example in Kwon et al (2019)'s paper.

2) Please explain in detail what part of the previous model you modified or proposed to improve the FCD performance.

This is an important point to clarify. We did not change the architecture of any of the baseline models. In order to measure correlations between metrics (Table 2) and plot the tradeoff between metrics (Figure 1 and Supplementary Figure 3) for the baseline models, we introduced a temperature hyperparameter in the sampling mechanism for the final softmax layer in each of these models. Changing the value of this temperature hyperparameter changes the sampling distribution, generating sets of molecules with different values of FCD, novelty, validity, and so on. We use a temperature hyperparameter because autoregressive models offer no other controllable generation hyperparameters.

3) I think it will be improved if we train the previous models (ex. NAT GraphVAE) with the FCD value as the target properties, what do you think?

Most existing approaches are based on maximum likelihood training i.e. they maximise the likelihood function of the data given the model parameters; they do not necessarily improve or maximise a particular metric. It is particularly difficult to maximise FCD as a training objective as it is a distributional metric which is calculated using the distribution of several molecules. Using it as a training objective would violate the independent and identically distributed assumption used to train deep learning models via likelihood maximisation. However, how to incorporate such metrics into training is definitely an interesting research direction for future work.

Reviewer #2 (Remarks to the Author):

Summary

First of all, I would like to give kudos to the authors on an extremely well-written paper. Many of the questions that I had as I was reading the paper were eventually answered in different areas of the Methods or SI, which I thought was very well done so that it does not disrupt the flow of the “main” text. Overall it was enjoyable to read this paper. Explanations were generally very clear (although I still have some questions which I outline below). I was also very happy to see that the code was provided and that it appeared to be clear and readable code. From a reproducibility standpoint, I think this manuscript would score very high as the authors did a good job of reporting what was done plus have made their code available.

Now to the science. In this work, the authors present a graph-based molecular generative model that operates by masking different elements on input graphs uniformly at random, learning the conditional distributions for the masked graphs, and then generating new graphs using their model. They refer to their approach as masked graph modeling. I believe the model is very interesting, although not perhaps so “powerful” as to say it competes with SOTA deep molecular generative models, given that the model can only learn to complete nearly complete graphs (correct me if I misunderstood). As such, it appears that the model would be quite interesting from a lead optimization point of view (if thinking about it from the context of drug discovery), but if, let’s say, the goal of the model was to find drug candidates with a completely different scaffold (e.g. more analogous to hit discovery) the model would be unable to do so (again, please correct me if I am wrong). As such, I think the authors could improve the impact of their method by “branding” it more as a tool for molecular optimization rather than as an outright de novo design tool.

As a deep molecular generative model, the model is ok. I am not impressed by the performance which in my opinion is very average, and I do not see this being implemented in e.g. a drug discovery pipeline, but I still find the approach very interesting. The authors also do a very interesting analysis. I have divided my feedback below into Major and Minor points.

We thank the reviewer for the comprehensive review covering all aspects of our work. We believe that reproducibility is essential in scientific work and hope others will be able to further interpret and build on our work by looking at our code and models in conjunction with the data that we use.

We would like to clarify that our generation procedure is not a one-step approach. Even though the training procedure consists of completing an incomplete graph, the generation procedure consists of repeatedly masking out and replacing randomly chosen parts of the same graph over many iterations. Following the principles of a denoising autoencoder [Alain and Bengio, 2014], which the masked graph model follows, sampling repeatedly from the trained model is equivalent to sampling from the full joint distribution. Hence our model can be used to generate

completely de-novo graphs, which will diverge from the original graph as the number of sampling iterations becomes large. For small numbers of sampling iterations, the molecules produced may well be similar to the original molecule, and it may be possible to use a further developed version of this approach for lead optimisation. The model can also be used for graph completion if only one sampling iteration is used at generation time. We leave a full exploration of these directions for future work.

Major Points

1) For starters, I thought it was a shame that the performance of the models significantly drops when masking more than 20% of the nodes, as my understanding is that being able to mask more nodes would possibly improve the diversity of the structures generated. Is this just a limitation of the underlying model used, or do you believe that greater masking percentages could be used by tuning the model (somehow)? Just curious.

Let us reiterate that our approach is not a single-step approach. We carry out the masking and reconstructing procedure multiple times. If we do this for a large enough number of iterations, then even with a small masking rate the model should produce diverse samples in accordance with the properties of a Gibbs sampler. However, if the number of sampling iterations is kept constant, a higher masking rate tends to lead to greater diversity. We can train models with a higher masking rate but we have to be careful not to corrupt the original graph so much that the model does not have enough signal to reconstruct the original graph from the part of the graph that remains. For example, to train the natural language BERT model, less than 15% of tokens are corrupted. Wang and Cho [2019] then use such a model to successfully carry out generation.

2) To clarify, in neither generation approach (i.e. using training or marginal initialization) can the model generate graphs starting from an empty graph, is this correct? No matter how the graphs are initialized for generation, they will have at most the unmasked % number of graph elements (nodes +edges). That is, if the mean # nodes in training set graphs is 30, and a 10% masking rate was used, then the graphs will be initialized with ~27 nodes? Looking for some clarification as this was the part that was not entirely clear in the text for me.

To follow up on our response to Question 1, the generation procedure is a multi-step procedure. So with enough sampling iterations we will replace all the graph components over multiple iterations. The graphs are initialised by either taking a training set graph or randomly initialising a graph with number of nodes drawn from the distribution of the size of training molecules. The sampling and replacing procedure is then carried out multiple times, with randomly chosen graph elements/features sampled and replaced each time.

As an example, for a graph with 30 nodes initialised using marginal initialisation and updated with a 10% masking rate: the graph is initialised with 30 nodes and some number of edges. At each iteration, each node feature is masked out at 3 different nodes and then updated using our model. This masking and updating procedure is carried out for several iterations. A similar analysis holds for edges (see next major point for more details), which are updated simultaneously with the nodes.

3) (this is related to point 2 above) I don't fully understand what 1% masking corresponds to in the models. To illustrate, I believe most molecular graphs in ChEMBL have between 20-50 heavy atoms (and let's say roughly 2 edges per node), which if we assume an average of 40 nodes per structures, means there are ~120 elements (nodes and or edges) in each graph to mask. So at 1% masking, does it mean each graph in the training set only has ~1.2 nodes or ~1.2 edges masked? On a related note, on pg. 14, the authors state "We recommend using a generation masking rate corresponding to masking out 5-10 edges of a complete graph having the median number of nodes in the dataset." but I don't see how 5-10 edges corresponds to 1-5% masking rate in ChEMBL and 10-20% in QM9. And once again, on pg. 17, the authors say "The number of edges masked and replaced for a median ChEMBL molecule with a 1% masking rate and for a median QM9 molecule with a 10% masking rate are both approximately 4." How does a 1% masking rate in ChEMBL corresponds to 4 edges, as there are not 400 edges on the typical ChEMBL molecule, but much fewer... Am I misunderstanding something? It could help to illustrate the masking process by including a scheme (maybe in the SI), though this is just a suggestion.

We have clarified this point in the main text (page 16). We now define masking in terms of nodes and 'prospective edges' where a prospective edge is defined as a possible edge between any two atoms in the graph. Therefore for a graph with N nodes, the number of prospective edges would be $N(N-1)/2$ when accounting for symmetry and the restriction that there are no self-loops. The masking rate is applied to this number, not to the 'actual' number of edges in the graph (as using the 'actual' number would be giving the model access to information it should not have during training, and for generation there is no 'correct' number of edges). We have included a schematic as Supplementary Figure 4 that contains an example for a graph initialised with 10 nodes.

4) I was wondering why a masking rate of 1-5% was used for ChEMBL and 10-20% for QM9? That is, it doesn't make sense to me that the performance of MGM would decrease faster with decreasing masking rate on ChEMBL, which has the larger structures, than on QM9. I think the authors tried to explain this in the text but it was not entirely clear the explanation.

We have further elaborated on this point in the main text (page 17). The masking rate is applied to the total number of 'prospective edges' in a graph, which scales as the square of the number of nodes. Keeping a constant masking rate would be problematic as the number of bonds in a molecule does not scale in this way and larger molecules in fact have sparser adjacency matrices. Having an abnormally large number of prospective edges masked out can cause problems for the MPNN as it will pass messages along these spurious edges between many parts of the graph that should have no connections between them. Finding a better heuristic or automated way of setting the masking rate would be a valuable direction for future research. As a note: 10-20% of the median number of nodes for QM9, and 1-5% for ChEMBL correspond to roughly the same absolute number of nodes.

5) As the model is carrying out iterative graph generation, it is not clear to me how/when the model knows to “stop” the graph generation? Similarly, in the graph generation process, how do you set how many nodes/edges the graph can have (i.e. the max value)? Can your model generate graphs larger than are in the training set? My understanding is not, as the model would not have seen anything like that before (should you initialize a larger graph than was seen in the training set), but again just looking for some clarification. A schematic for the graph generation process could be useful here.

It is an open question as to when to stop sampling during generation. As the number of sampling iterations increases, the distribution from which the samples is taken remains the same, therefore there is no need to stop at any specific point. When to start/stop sampling from a Markov Chain Monte Carlo sampler is an ongoing area of research.

During graph generation, for a particular graph, the number of nodes in the graph is fixed (though it may be possible to modify the framework to alleviate this restriction in future work).

For training initialisation, the number of nodes is equal to the number of nodes in the initial training set graph. For marginal initialisation, we look at the marginal distribution over graph sizes in the training set. So for example, if 80% of graphs in the training set have 9 nodes, there is an 80% chance that a newly initialised graph will have 9 nodes.

Our model should be able to generate graphs that are larger than those in the training set i.e. the number of nodes in the initialised graph can be set to be higher than any graph in the training set. Since the MPNN in our model is invariant to the size of the graph and is only concerned with a node's nearby receptive field, it should be possible to carry out the generation process. It is important when doing this to keep in mind our comments about the effect of the masking rate as the graph size increases.

We carried out a preliminary generation experiment by training a model on a subset of the ChEMBL dataset containing molecules with at most 50 nodes, and generated molecules with up to 88 nodes. We found that this did not change the values of the evaluation metrics by much. We provide a schematic for the graph generation process in Supplementary Figure 4.

6) Which initialization method is used for the results shown in Table 1 & Figure 1 (training or marginal)? Also, would they lead to different results?

We use both training and marginal initialisations for Table 1 and Figure 1. For Figure 1, the MGM points originating at the bottom right correspond to marginal initialisation whereas those originating at the top left correspond to training initialisation. We have clarified this in the text. The results are different as the novelty is 0 and FCD is ~1 when starting with training initialisation, whereas the reverse is true for marginal initialisation. Both initialisations seem to converge at roughly the same values of metrics for QM9. For ChEMBL, only training initialization was used because marginal initialisation did not yield enough valid molecules to calculate reliable distributional metrics in a reasonable amount of time. This is likely because the masking rate is low so it would take a very long time for the sampler to converge. With a high masking rate we run into the problem of too many spurious edges as outlined in the response to major point 4. Finding a way to alleviate this issue would be a valuable direction for future work. We have added this explanation to the caption for Figure 1.

7) In Table 4, the authors compare to other SOTA models, and say their “masked graph model corresponds to 1% masking rate and training graph initialization”. If I understand correctly, then this means the authors start generation from a nearly complete graph and add the remaining 1% of nodes/edges. I am guessing that, in comparison, the LSTM is starting from an “empty” molecular string, adding tokens to the string until sampling the termination token. So, a more “fair” comparison for MGM against the LSTM would be to start from the SMILES corresponding to the “masked” training set structure and letting the model complete them, no? I suppose the autoencoder and GAN models used in Figure 4 generate new structures in a “one-shot” manner. For the Graph MCTS, I believe it also starts from molecular fragments. So comparing to the GAN/VAE/MCTS methods makes “sense”, while comparison to the LSTM is a bit like comparing apples and oranges. Can you comment a bit on this? I am not suggesting to remove comparison to the LSTM, as the LSTM has SOTA performance for some generation tasks and this is obviously an interesting comparison, but just to discuss a bit more what is being compared.

After our clarifications regarding the multi-step nature of our generation process, we hope that the comparison with the LSTM seems more suitable. As noted earlier in our responses, due to the multiple iterations of masking and replacing components with MGM, we are in effect sampling from the full joint distribution rather than completing an initial graph. If we were to use only one sampling iteration instead, then a comparison with an LSTM that started with a partially completed molecule could be more suitable. In summary, an LSTM and an MGM both learn the joint distribution but in different ways. To sample from the joint distribution, the parameterisation of an LSTM necessitates the need to start generation from an empty string, whereas the parameterisation of MGM lends itself to an approach of repeatedly masking out and replacing parts of a graph.

8) I would recommend part of the background in Appendix A to be discussed in the main text, namely a bit more background on graph-based generative models. I say this because on pg. 9 (and other places throughout the manuscript), the authors claim that their approach “outperforms” existing graph-based methods, which I would say is a bit of an overstatement, given that the model was only compared to “old” graph-based generative models. While models like MolGAN are definitely good “baselines”, it is definitely not SOTA. So I believe it is important to mention that there are plenty of other graph-based generative models which have been since published (e.g. DEFactor by Assouel et al (2018), MolGAN by De Cao et al (2018), RVAE by Ma et al (2018), GraphVAE by Simonovsky et al (2018), GraphNVP by Madhawa et al (2019), MoFlow by Zang et al (2020), NeVAE by Samanta et al (2018), MolRNN by Li et al (2018), JTN-VAE by Jin et al (2018), some models by DeepMind, Li et al (2018), HierVAE by Jin et al (2020), GraphINVENT by Mercado et al (2020), GCPN by You et al (2018), GraphAF by Shi et al (2020), MolecularRNN by Popova et al (2019), to give just a few examples; I saw after I wrote this that many of these were already discussed in the SI, which was great, maybe the authors could pick a few of the better performing graph-based generative models to discuss in the main text as well). Given that this is a

publication on a graph-based molecular generative model, I think it is appropriate that the authors expand a bit here, and provide more background to the reader.

Thank you for the extensive list of relevant works provided. We have added new methods to the main text that we had not previously cited in either the main text or the Supplementary Information. We have elaborated on some of the methods we had cited. We have also moved some of the related work in the Supplementary Information section to the main text. We would also like to note that these papers use varying evaluation criteria and datasets.

9) Curious how the model performance changes when fewer/additional atom features are included (if this was explored at all)?

This was explored and is now part of the text (page 19). Using fewer features led to a significantly higher validation loss.

10) Also curious how long a typical training run takes for MGM? What about a generation run? Let's say, for the QM9 dataset and on whichever GPU you used. I'm just trying to get an idea for what the computational cost is for the model, because I assume it is relatively "cheap" for a graph-based generative model - and if this is the case I believe you can (and should) "sell" this aspect more in your manuscript. If you have numbers for the specific training time per epoch that would be interesting to see, too.

We have now included information on training and generation times for each dataset in the results section (page 12 and Table 6).

11) On pg. 14, the authors say "Developing datasets and benchmarks that incorporate graph-level information that is not readily encoded as strings, such as spatial information, would alleviate this issue." yet there do exist other benchmarks (e.g. molecular rediscovery benchmarks, which are actually implemented in GuacaMol and it is unclear why the authors did not use them; also benchmarks related to the coverage of chemical space, see Arús-Pous et al (2019)

<https://jcheminf.biomedcentral.com/articles/10.1186/s13321-019-0393-0>). So this statement should be reworded, as other benchmarks do exist, so are these benchmarks not sufficient to address the string vs graph comparison (and if so, why not?), or are the authors planning on developing another benchmark anyhow?

We have rephrased this and cited the Arús-Pous et al benchmark to clarify that even though other benchmarks such as this exist, they use string representations of molecules. So they do not evaluate molecules for graph-based properties such as spatial geometry, which text-based models cannot represent or output. Furthermore, if the metric that the reviewer is referencing is the UC-JSD metric used in the Arús-Pous et al paper, we would note that this metric is only applicable to models that explicitly parameterise the joint distribution. It is however an interesting future research direction to study how to come up with such a metric for models that implicitly parameterise the distribution such as MGM.

12) On pg. 14, there is this statement: "Future avenues of work include incorporating additional information such as inter-atomic distances into our graph representations. The development of benchmarks that account for the graphical representations of molecules,

for example by incorporating spatial information, would help more fairly compare graph-based generative models to each other and to SMILES based models.” I disagree with this statement, actually. Since the string-based representation is built on the graph-representation, I believe the string- and graph-based generative models (for 2D molecules) are entirely comparable if they have used the same node- and edge-features in their respective representations. So I would recommend to the authors to rephrase this statement (or if not, let me know why you disagree with me). Nonetheless, I do think generation of 3D structures from either string- or graph-based representations is extremely interesting!

We have rephrased our statement to emphasise that there are graph-level features that are not readily encoded as strings. Even though string-based and graph-based models may be comparable when considering certain 2D representations, graph-based methods allow for the incorporation of features specific to 3D representations, such as spatial information.

String-based benchmarks do not use or evaluate such information. In the GuacaMol benchmark, for example, since the data is provided as strings and must be converted back into strings for evaluation, these features are not used by either the text-based or graph-based models and cannot be a part of evaluation.

Furthermore, even though it is possible to linearise graphs into strings, they can pose long range dependency problems to neural networks. Graph-based representations may be more efficient in capturing the properties of a graph as these dependencies are explicitly modelled by edges.

13) On pg. 17, the authors say “We use more layers for ChEMBL because more message passing iterations are needed to cover a larger graph.” Was this determined through some kind of hyperparameter optimization procedure or is this just a “guess”? From what I have seen in the molecular property prediction and molecular generation literature, the number of “optimal” message passes in an MPNN does not necessarily correlate with the size of the graphs in the training set. One could also make the argument that if the model is searching for certain “pharmacophores” (e.g. in toxicity prediction), even if training set graphs are large the optimal number of message passes might still be small for toxicity prediction. So what I am saying is that I would like more details on how the authors determined the optimal number of message passes, even if they just chose these numbers based on intuition.

We now provide Supplementary Table 2, which details the results of a comprehensive hyperparameter search on ChEMBL.

Minor Points

14) In Figure 1, why is the correlation not as smooth for the QM9 dataset as when the model was trained on ChEMBL? That is, why the bump in the MGM curves (for both masking %) in Figure 1a? Maybe it is hard to know why, but just curious if you had an idea.

As we mentioned earlier in our responses and have added to the text, QM9 results in Figure 1 correspond to training and marginal initialisation whereas ChEMBL results correspond to

training initialisation only. The part of the QM9 plot where the correlation is not so smooth is where points from training and marginal initialisation converge after several iterations.

15) I like Table 1, showing the correlation between different Guacamol metrics. Based on these correlations, the authors suggest that they can look at a subset of the metrics, namely the uniqueness, Fréchet and novelty scores, to gauge generation quality. I thought this was really valuable.

Thank you, we agree that it is useful to see how the metrics are correlated and use this information in gauging generation quality.

16) I also like the idea that the smoother change in the Fréchet ChemNet Distance score vs the novelty allows the authors a greater degree of control in MGM compared to other molecular generative models. I found this an interesting metric and a cool angle from which to look at molecular generative model performance.

Thank you, we agree that looking at the tradeoff is useful compared to simply summing or multiplying metrics, which presents problems when the metrics are correlated.

17) Were heavy atoms only considered as nodes, or hydrogens as well? (the list on pg. 16 makes me think only heavy atoms but just wanted to confirm)

You are correct, only heavy atoms are considered as nodes. We have now made this explicitly clear in the text (page 15).

18) Some of the molecules in Figure 3 are just crazy (from a chemistry standpoint). My recommendation is to pick another example to show for e.g. Figure 3a (esp after 400 steps). I personally don't care what example is shown (and I actually like this example, I find it very useful from the perspective of deep generative models), but since this is not a dedicated computational journal, I think maybe it is worth updating the example you show. If a chemist were to see these "examples", my guess is they would not be inclined to use your model. Of course, your goal may not be to convince a chemist to use your model per se (at least not at the moment), so this is highly subjective advice...

The examples were chosen to show how molecules change over the course of sampling iterations. For further reference, we now provide figures in the supplementary material corresponding to novel molecules with relatively high QED scores. We also now provide access to full lists of SMILES strings obtained after the final sampling iteration in the generation process via the GitHub repository.

19) In pg. 16, how were these properties computed for the molecules in the training set, from looking at the code I figured RDKit but will be good to mention it in the paper too.

Yes, these properties were computed using RDKit. We have now mentioned this in the text (page 15).

20) On pg. 14, the authors say "There are several differences between the QM9 and ChEMBL datasets that could account for this, including number of molecules, median

molecule size and presence of chirality information.” Here you could point the reader to the section in the Methods, since there you describe what the specific differences are.

We have now done this.

21) Just a suggestion, but for the GitHub repo, I recommend to include an environment file so users can easily install the *exact* same dependencies on their system as you have used. Even sometimes installing packages in a slightly different order leads to slightly different dependencies, and I have seen this lead to problems (especially when PyTorch, tensorflow, and RDKit are involved).

We have now included an environment.yml file in the GitHub repo and referenced it in the README.

Reviewer #3 (Remarks to the Author):

The generation of novel and sensible molecular structures is a key problem in computer aided drug discovery. The authors describe a novel approach for molecule generation based on masked graph models (MGM), which is complementary to previous approaches based on string representations or graph representations. Furthermore, results on the conditional generation of molecules are presented, where molecules with desirable molecular property ranges are generated.

Furthermore the authors perform a critical analysis of the recently proposed guacamol metrics, showing that some are correlated, and perform a multi-criterial assessment of the models to highlight that MGM can be tuned well to populate the full pareto front, whereas SMILES-based models are not as easy to control.

The paper also has interesting results on SMILES transformers, which underperform SMILES-LSTM models, which contrasts recent results in machine translation and NLP.

Overall this is a valuable contribution, which I would recommend for publication after the following points have been addressed.

We thank the reviewer for their feedback. We answer the questions raised below.

Questions:

Is it really a surprise that novelty and frechet distance (FCD) are negatively correlated? After all, the FCD measures the distance between the sampled molecules to a set of known molecules, whereas novelty measures how
Note that in the context of molecular design novelty by itself is of limited value, since we can trivially generate new molecules with unnatural structures.

We agree that it makes sense intuitively that these two metrics are negatively correlated. However, we have not seen previous work that points this out or discusses it, and in fact we have seen instances where models are evaluated by criteria such as calculating the geometric mean of the metrics for each model. Such comparisons are misleading if the metrics are correlated, so we think it is important for researchers to be aware of the correlation and take it into account when making comparisons.

Missing citations:

It would be good cite the original paper which introduced autoregressive LM type models for molecules (SMILES-LSTM) <https://arxiv.org/abs/1701.01329>

We have now included this citation in the main text.

**Additionally, I would recommend to cite work on reaction-driven molecule design, where molecules are generated by performing virtual chemical reactions, e.g.
<https://arxiv.org/abs/2012.11522>**

We have now included this, and other relevant citations on reaction-driven molecule design, in the Introduction section of the main text.

REVIEWERS' COMMENTS

Reviewer #1 (Remarks to the Author):

I have confirmed the authors' reasonable response and the revised manuscript. I had additional questions, which were explained by the author's answers to the questions in Reviewer #2. However one question remains: I wonder if your proposed method is possible to train and generate novel molecules using training data with more than 100 heavy atomic sizes in molecular structures. In many cases, the novel molecular structure generation models does not consider molecular size because they have targeted drug. However, using the graph type representations requires a lot of training time and memory depending the molecular size. Please explain how much molecular size you can train and generate with the proposed method. If possible, explain the number of heavy atoms in a molecules.

Reviewer #2 (Remarks to the Author):

Thank you to the authors for their thorough revisions to the manuscript. The authors have addressed all my comments, and I believe the manuscript is even clearer now. I like the new additions both to the main text and SI. I have no further comments now and would like to recommend the manuscript for publication. This work would be highly interesting to others working in the field of deep molecular generative models.

Reviewer #1 (Remarks to the Author):

I have confirmed the authors' reasonable response and the revised manuscript. I had additional questions, which were explained by the author's answers to the questions in Reviewer #2. However one question remains: I wonder if your proposed method is possible to train and generate novel molecules using training data with more than 100 heavy atomic sizes in molecular structures. In many cases, the novel molecular structure generation models does not consider molecular size because they have targeted drug. However, using the graph type representations requires a lot of training time and memory depending the molecular size. Please explain how much molecular size you can train and generate with the proposed method. If possible, explain the number of heavy atoms in a molecules.

We thank the reviewer for looking over our responses and revisions. To answer the proposed question, we first state the definition of a heavy atom as any atom in the molecule that is not a hydrogen atom. It is possible to train on a dataset with molecules of size greater than 100 heavy atoms, and then use the trained model to generate molecules with size greater than 100 heavy atoms. The ChEMBL dataset comes close with a maximum molecule size of 88 heavy atoms. We would like to reemphasise the point we made in our revision to the manuscript, that there may be issues with using a constant masking rate as the molecule size becomes very large, because the number of prospective edges in a molecule scales faster than the number of actual edges. This could cause problems for the type of message passing neural network architecture that we use. Possible ways of alleviating this issue include coming up with more sophisticated masking schemes, or using/developing model architectures that discourage message passing along spurious prospective edges. These are all interesting directions for future work.

Reviewer #2 (Remarks to the Author):

Thank you to the authors for their thorough revisions to the manuscript. The authors have addressed all my comments, and I believe the manuscript is even clearer now. I like the new additions both to the main text and SI. I have no further comments now and would like to recommend the manuscript for publication. This work would be highly interesting to others working in the field of deep molecular generative models.

We once again thank the reviewer for their comprehensive review and are happy to hear that our revisions enhanced the clarity of the manuscript.